**Letter**

# Genetic risk factors for COVID-19 and influenza are largely distinct

Coronavirus disease 2019 (COVID-19) and influenza are respiratory illnesses caused by the severe acute respiratory syndrome coronavirus 2 (SARS-CoV-2) and influenza viruses, respectively. Both diseases share symptoms and clinical risk factors[1], but the extent to which these conditions have a common genetic etiology is unknown. This is partly because host genetic risk factors are well characterized for COVID-19 but not for influenza, with the largest published genome-wide association studies for these conditions including >2 million individuals[2] and about 1,000 individuals[3–6], respectively. Shared genetic risk factors could point to targets to prevent or treat both infections. Through a genetic study of 18,334 cases with a positive test for influenza and 276,295 controls, we show that published COVID-19 risk variants are not associated with influenza. Furthermore, we discovered and replicated an association between influenza infection and noncoding variants in *B3GALT5* and *ST6GAL1*, neither of which was associated with COVID-19. In vitro small interfering RNA knockdown of ST6GAL1—an enzyme that adds sialic acid to the cell surface, which is used for viral entry—reduced influenza infectivity by 57%. These results mirror the observation that variants that downregulate *ACE2*, the SARS-CoV-2 receptor, protect against COVID-19 (ref. 7). Collectively, these findings highlight downregulation of key cell surface receptors used for viral entry as treatment opportunities to prevent COVID-19 and influenza.

To understand the extent to which the same host genetic factors influence the risk of coronavirus disease 2019 (COVID-19) and influenza, we first performed a genome-wide association study (GWAS) of influenza infection based on survey data from 296,313 participants of the AncestryDNA COVID-19 study who consented to the research[8]. Although the focus of that study was on risk factors for COVID-19, participants also indicated if they were tested for influenza in either the 2019–2020 or 2020–2021 flu seasons (Methods). Overall, 18,448 (6.2%) participants reported a positive test for influenza, and thus were considered cases for our analysis, while the remaining 277,865 participants (including 23,985 with a negative test) were considered population-level controls. We refer to this phenotype as 'reported influenza infection', but recognize that it does not represent true susceptibility to infection because the control group includes an undetermined number of individuals not exposed to influenza in either season or who were infected but not tested (for example, asymptomatic). As such, this phenotype may

capture symptomatic influenza infection that required seeking (or being prescribed) a viral test.

Using these data from AncestryDNA, we tested the association between reported influenza infection and 10 million common (frequency >1%) imputed variants using REGENIE[9], separately in three ancestral groups (with >100 influenza cases) defined based on genetic similarity to three superpopulations studied by the 1000 Genomes Project[10] (Methods): from Europe (EUR; *n* = 254,750, 86.0%), Africa (AFR; *n* = 12,951, 4.4%) and the Americas (AMR; *n* = 26,928, 9.1%), totaling 18,334 cases and 276,295 controls (Supplementary Table 1). Results were meta-analyzed across ancestries using an inverse-variance, fixed-effects meta-analysis (Extended Data Fig. 1), identifying two loci associated with reported influenza infection at *P* < 5 × 10⁻⁸ (near *ST6GAL1* and *B3GALT5*, respectively on chromosomes 21q22.2 and 3q27.3; Table 1). We describe these loci in detail later, including sensitivity and replication analyses in independent cohorts that demonstrate the reproducibility of these associations.

✉e-mail: goncalo.abecasis@regeneron.com; manuel.ferreira@regeneron.com

**Table 1 | Two loci identified in a multiancestry meta-analysis of reported influenza infection performed in the AncestryDNA cohort[a] and validated in an independent meta-analysis consisting of seven biobanks with electronic medical records[b]**

| Variant (effect allele) | Analysis | OR (95% CI) | P | Effect allele frequency in cases/controls | Homozygote OR (95% CI) |
|---|---|---|---|---|---|
| Chromosome 3q27.3, nearest gene *ST6GAL1* | | | | | |
| rs16861415 (C) | Discovery | 0.864 (0.826–0.903) | $1.4 \times 10^{-10}$ | 0.064/0.074 | 0.627 (0.489–0.804) |
| | Replication | 0.901 (0.872–0.930) | $3.3 \times 10^{-10}$ | 0.094/0.097 | 0.802 (0.717–0.894) |
| Chromosome 21q22.2, nearest gene *B3GALT5* | | | | | |
| rs2837112 (A) | Discovery | 0.901 (0.882–0.922) | $1.3 \times 10^{-19}$ | 0.460/0.485 | 0.824 (0.789–0.860) |
| | Replication | 0.936 (0.917–0.954) | $4.1 \times 10^{-11}$ | 0.432/0.461 | 0.877 (0.843–0.913) |

[a]18,334 cases versus 276,295 controls in the discovery analysis. [b]UK Biobank (UKB), GHS, PMBB, CCPM, Mayo Clinic, UCLA and FinnGen consisting of 22,022 cases and 1,131,269 controls in the replication analysis. Unadjusted *P* values were derived using Firth regression (two-sided test) as implemented in REGENIE[9].

Results from the AncestryDNA GWAS of reported influenza infection were then used to determine if severe acute respiratory syndrome coronavirus 2 (SARS-CoV-2) and influenza infections have a shared genetic etiology. To address this question, we initially focused on 24 variants associated with COVID-19 identified by the Host Genetics Initiative (HGI)[2] (freeze 6; Supplementary Table 2). Of these, only one was associated with reported influenza infection (*P* < 0.05/24 = 0.002), despite adequate power for most (Supplementary Table 3): rs505922 in *ABO* (odds ratio (OR) = 1.05 for the T allele, 95% confidence interval (CI) = 1.02–1.07, *P* = $2.2 \times 10^{-4}$; heterogeneity test *P* = 0.13; Fig. 1). This variant increased the risk of reported influenza infection, while it decreased the risk of COVID-19 (OR = 0.92; 95% CI = 0.92–0.93, based on the HGI GWAS of reported infection[2]), in line with previous reports[11]. We explore the *ABO* locus in greater detail in the Supplementary Note, concluding that its association with influenza is (1) only partially attenuated after accounting for COVID-19 status and (2) probably tags an underlying causal variant shared with other diseases (for example, childhood ear infections, allergic disease) but not COVID-19. Overall, only 10 (42%) of 24 variants had a consistent direction of effect on both influenza and COVID-19 (Fig. 1).

The lack of significant and directionally consistent associations between reported influenza infection and COVID-19 loci suggests that the two diseases share few—if any—genetic risk factors. Consistent with these findings, the two risk variants for reported influenza identified in the AncestryDNA GWAS (in or near *B3GALT5* and *ST6GAL1*) did not have a directionally consistent association with COVID-19 in the HGI analysis (Supplementary Table 4). Furthermore, the genetic correlation ($r_g$)[12] between reported influenza infection and both SARS-CoV-2 infection ($r_g$ = 0.30, *P* = 0.009) and COVID-19 hospitalization ($r_g$ = 0.34, *P* = 0.007) was modest (Supplementary Table 5). Collectively, these results suggest some sharing, but substantial divergence, in the genetic etiology underpinning influenza infection and COVID-19.

The AncestryDNA GWAS of reported influenza infection identified two associated loci (Table 1), with lead variants rs16861415 in *ST6GAL1* (3q27.3; OR = 0.86 for C allele, 95% CI = 0.83–0.90, *P* = $1.4 \times 10^{-10}$) and rs2837112 in *B3GALT5* (21q22.2; OR = 0.90 for A allele, 95% CI = 0.88–0.92, *P* = $1.3 \times 10^{-19}$). The effect allele ranged in frequency between 3% (AFR) and 8% (EUR) for rs16861415, and between 39% (AFR) and 49% (EUR) for rs2837112, with no evidence for heterogeneity of effect sizes across ancestries or cohorts (Supplementary Table 6). The reduction in influenza risk observed in homozygous carriers was 37% for *ST6GAL1* and 20% for *B3GALT5* (Table 1), with no evidence for epistasis between the two loci (Supplementary Note).

Next, we performed sensitivity and replication analyses to determine if the two influenza associations were robust to phenotype definition and reproducible. In the AncestryDNA cohort, excluding 253,880 individuals without influenza test results from the control group (resulting in 18,448 positive test cases versus 23,985 negative test controls) did not impact the effect size estimate for either locus: OR = 0.86 (versus 0.86) and *P* = $5.2 \times 10^{-6}$ for *ST6GAL1*, and OR = 0.89 (versus 0.90) and *P* = $4.9 \times 10^{-12}$ for *B3GALT5* (Fig. 2). In contrast, defining influenza infection more loosely based on whether a participant reported having flu-like symptoms in the 2019–2020 or 2020–2021 flu seasons (43,956 cases versus 250,673 controls) led to attenuated effect sizes but still highly significant associations: OR = 0.93 and *P* = $1.7 \times 10^{-7}$ for *ST6GAL1*, and OR = 0.95 and *P* = $4.0 \times 10^{-11}$ for *B3GALT5* (Fig. 2).

To determine if the associations were reproducible, we used data from medical records to define lifetime influenza infection status across 1,153,291 individuals from seven biobanks and five ancestral groups (Methods). Based on the presence of International Statistical Classification of Diseases and Related Health Problems, 10th Revision (ICD-10) codes J09, J10 or J11 in hospital admissions, general practitioner records or death registries, we identified 22,022 (2%) individuals with (cases) and 1,131,269 without (controls) a lifetime medical record of influenza (Supplementary Table 1). As with the AncestryDNA GWAS, the control group in this replication analysis probably includes both individuals not exposed to influenza and individuals who had influenza but not an associated medical record. In a multiancestry meta-analysis of medical record influenza (Extended Data Fig. 2), we observed directionally consistent and genome-wide significant associations with both rs16861415 in *ST6GAL1* (OR = 0.90, *P* = $3.0 \times 10^{-10}$) and rs2837112 in *B3GALT5* (OR = 0.93, *P* = $2.5 \times 10^{-11}$; Table 1). Two measures of recent influenza infection also supported both associations. First, we found consistent and significant associations with a positive culture for influenza A (Methods), an indicator of current infection available in 82,348 individuals from the Geisinger Health Study (GHS) biobank: OR = 0.82 and *P* = 0.005 for *ST6GAL1*, OR = 0.86 and *P* = $3.02 \times 10^{-5}$ for *B3GALT5* (Fig. 2). Second, both variants lowered the risk of a positive seropositive test for influenza A in a published study of 1,000 individuals[6], significantly so for *B3GALT5* (OR = 0.70, *P* = 0.001; Fig. 2). Lastly, the *B3GALT5* variant significantly lowered the risk of flu-related hospitalization among influenza cases (1,696 hospitalized cases versus 8,239 nonhospitalized cases, OR = 0.88, *P* = 0.005), with a similar, albeit nonsignificant, protective effect for the *ST6GAL1* variant (OR = 0.89, *P* = 0.17; Fig. 2). Collectively, these findings establish both loci as reproducible genetic risk factors for influenza and indicate that the *B3GALT5* variant also reduces disease severity.

We did not find any additional associated loci in the meta-analysis of discovery (AncestryDNA) and replication (biobank) cohorts (40,356 cases versus 1,407,564 controls; Extended Data Fig. 3). As observed in the AncestryDNA GWAS, aside from *ABO*, published COVID-19 risk variants were not associated with influenza in this larger analysis (Extended Data Fig. 4).

Next, to help understand how each influenza locus contributes to disease pathophysiology, we identified the likely effector genes of the GWAS signal, concentrating on the lead variant at the 3q27.3

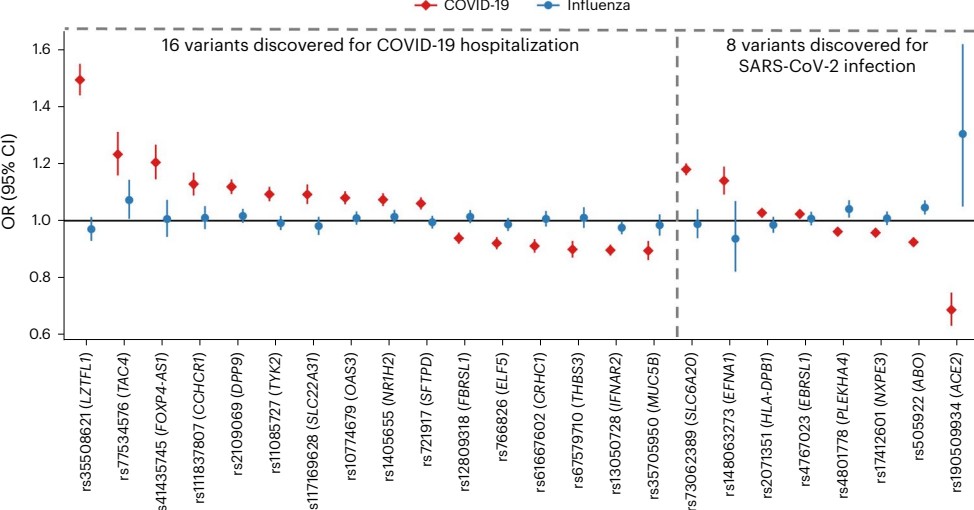

**Fig. 1 | Association between reported influenza infection in the AncestryDNA cohort and 24 variants previously reported to be associated with COVID-19 outcomes by the HGI.** Of the 24 COVID-19 risk variants, 16 were discovered in a GWAS of COVID-19 hospitalization (comparing 25,027 cases hospitalized with COVID-19 against 2,836,272 individuals with no record of SARS-CoV-2 infection), while eight were discovered in a GWAS of reported SARS-CoV-2 infection (comparing 125,415 individuals with a record of SARS-CoV-2 infection against 2,575,157 individuals with no record of SARS-CoV-2 infection). Of the 24 variants, only one (rs505922, 9:133273813:C:T, in *ABO*), was associated with reported influenza (18,334 cases versus 276,295 controls) after Bonferroni correction for 24 tests (*P* = 0.002, obtained using Firth regression, two-sided test); however, the direction of effect for influenza (blue circles) was the opposite of that reported for COVID-19 (red diamonds). The error bars represent the 95% CI for the OR estimate.

and 21q22.2 loci in the meta-analysis of the discovery and replication cohorts, that is, rs13322149 and rs2837113, respectively (Extended Data Fig. 3). Based on high linkage disequilibrium (LD, $r^2 > 0.80$) between each variant, and sentinel expression quantitative trait loci (eQTLs) and enhancer-overlapping variants (Supplementary Tables 7 and 8), four genes were prioritized: *ST6GAL1* and *ADIPOQ* at the 3q27.3 locus; and *B3GALT5* and *IGSF5* at the 21q22.2 locus. Analysis of rare loss-of-function (LOF) and missense variants assayed via exome sequencing of 14,189 cases with influenza and 811,714 controls did not identify any significant genome-wide associations (Extended Data Fig. 5); however, when we focused on the four genes highlighted above, we found a missense variant in *IGSF5* (frequency 0.01%) associated with a 9.2-fold higher risk of medical record influenza, which was significant after correcting for 631 rare variant tests performed across the four genes ($P = 2.3 \times 10^{-5}$; Supplementary Table 9). This observation provides additional support for *IGSF5* as one of the likely effector genes underlying the common variant association with flu at the 21q22.2 locus.

Of the four likely effector genes of the influenza loci, *ST6GAL1* and *B3GALT5* are strong biological candidates (*ADIPOQ* and *IGSF5* are discussed in the Supplementary Note). *ST6GAL1* codes for the enzyme β-galactoside α-2,6-sialyltransferase 1, which catalyzes the addition of sialic acid to galactose by an α-2,6 linkage[13]; it is most highly expressed in the liver and in Epstein–Barr virus-transformed B cells in humans (Extended Data Fig. 6a)[14]. Critically, influenza virus infection is initiated when the viral hemagglutinin glycoprotein binds to an α-2,6-linked sialic acid found on human host cell surface glycoproteins and glycolipids in the upper respiratory tract, which are used by the virus as attachment factors that facilitate the subsequent engagement with a functional receptor required to enter the target cell[15–17]. The lead variant at this locus (rs13322149) colocalized with a sentinel eQTL (rs73187789:A, $r^2 = 0.95$) that is associated with lower expression of *ST6GAL1* in thyroid tissue from the Genotype-Tissue Expression (GTEx) project[14] ($P = 3.4 \times 10^{-12}$; Supplementary Table 7), with consistent directional effects in other tissues, including the lung (Extended Data Fig. 6b). *B3GALT5* codes for β-1,3-galactosyltransferase 5 and is most highly expressed in the small intestine and salivary gland (Extended Data Fig. 6c)[14]. This enzyme catalyzes the addition of galactose in the β-1,3 conformation to an *N*-acetylglucosamine (GlcNAc) saccharide

during the synthesis of glycan core structures[18]. As noted above, *ST6GAL1* adds sialic acid to a galactose. The lead variant at this locus (rs2837113) is a sentinel eQTL for *B3GALT5* in skin and salivary gland tissue, with the rs2837113:A influenza-protective allele associating with higher gene expression (Supplementary Table 7).

Lastly, we performed in vitro experiments to study the impact of gene expression knockdown on influenza virus H1N1 (Puerto Rico 8 strain) infectivity. For these experiments, we focused on two likely effector genes of influenza-associated variants—*ST6GAL1* and *B3GALT5*—because of their potential role in a critical step of influenza virus infectivity, that is, modulation of α-2,6-linked sialic acid abundance at the cell surface. We tested two small interfering RNAs (siRNAs) per gene in the A549 and Calu-3 cell lines, respectively, performing two independent experiments per siRNA. siRNAs against *ST6GAL1* achieved approximately 90% expression knockdown and resulted in approximately 80% reduction in sialic acid abundance at the cell surface and approximately 50% reduction in influenza infectivity (Extended Data Figs. 7 and 8), which is consistent with previous findings[19]. These results support the notion that lower ST6GAL1 enzymatic activity reduces the ability of influenza virus to infect host cells, a mechanism that probably explains the association between variants at the 3q27.3 locus and lower risk of influenza infection. In contrast, knockdown of *B3GALT5* expression was not associated with a consistent effect on influenza infectivity (Extended Data Fig. 9). As such, despite being a good biological candidate, it is unclear if *B3GALT5* underlies the association at the 21q22.2 locus.

There are several important limitations that should be considered when interpreting the results from this study (discussed in detail in the Supplementary Note), including (1) phenotype misclassification, (2) potential confounding effects of unmeasured risk factors for influenza infection, (3) the use of self-reported influenza information in the AncestryDNA cohort; and (4) an undetermined influenza strain infecting GWAS participants.

In conclusion, we demonstrated that the genetic architectures of COVID-19 and influenza are mostly distinct, with few shared common genetic risk factors. We identified and replicated the first genome-wide-significant loci for influenza and demonstrated that inhibition of *ST6GAL1* reduces viral infectivity in vitro. Host genetic studies of infectious diseases commonly identify protective variants

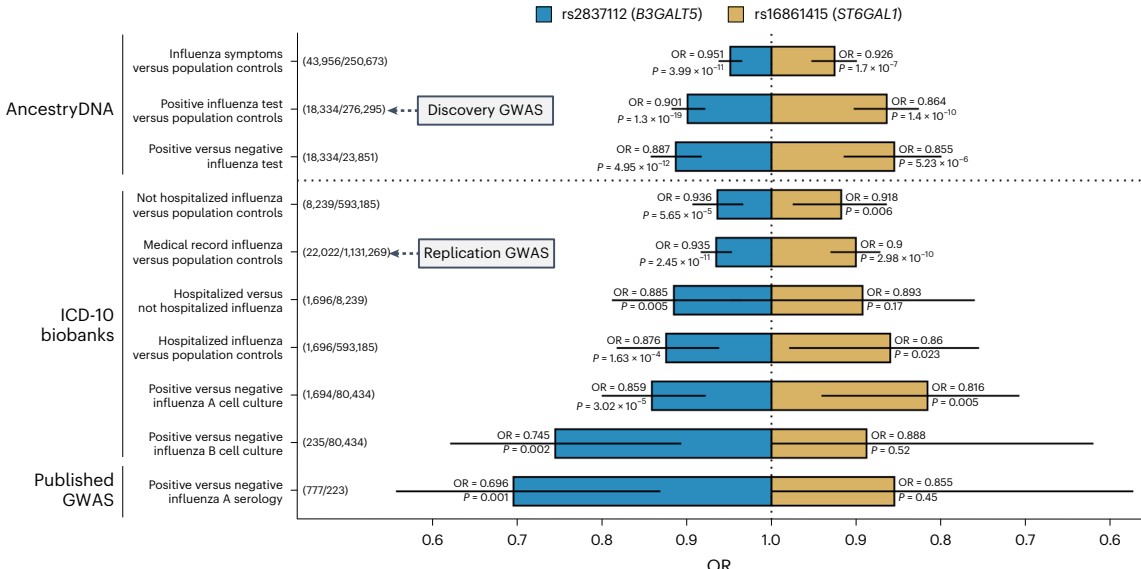

**Fig. 2 | Association between *ST6GAL1* and *B3GALT5* variants and ten influenza-related phenotypes.** Variants in *ST6GAL1* (rs16861415) and *B3GALT5* (rs2837112) were associated with lower risk of reporting a positive test for influenza in the AncestryDNA cohort (discovery GWAS, total *n* = 294,629). The association between both variants and influenza infection was confirmed when analyzing medical record-based influenza status in an independent analysis of 1,153,291 individuals from seven biobank cohorts (replication GWAS; the cohorts are listed in Supplementary Table 1). Sensitivity analyses based on eight additional phenotypes showed that (1) in the AncestryDNA cohort, effect sizes for both variants were comparable after excluding controls with no available influenza test results, while they were weaker when testing a looser

phenotype that considered only flu-like symptoms; (2) in two biobank cohorts with available hospitalization data (UKB, GHS), restricting the case group to individuals with influenza-related hospitalization resulted in stronger effect sizes for both variants, with the *B3GALT5* variant significantly reducing the risk of hospitalization among infected cases; and (3) consistent and often stronger (by effect size) associations were observed with phenotypes that captured recent influenza infection, such as a positive cell culture or serology test for influenza A. Further details on these associations are provided in Supplementary Table 6. Unadjusted *P* values were derived using Firth regression (two-sided test) implemented in REGENIE[9]. The error bars represent the 95% CI for the OR estimate.

that putatively downregulate (or ablate) proteins required for viral entry (*CCR5* in HIV[20], *ACE2* in SARS-CoV-2 (ref. 7) and *FUT2* in noroviruses[21]). Our findings provide the latest vignette to this evolving narrative.

## Online content

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

Jack A. Kosmicki[1], Anthony Marcketta[1], Deepika Sharma[1], Silvio Alessandro Di Gioia[1], Samantha Batista[1], Xiao-Man Yang[1], Gannie Tzoneva[1], Hector Martinez[1], Carlo Sidore[1], Michael D. Kessler[1], Julie E. Horowitz[1], Genevieve H. L. Roberts[2], Anne E. Justice [3], Nilanjana Banerjee[1], Marie V. Coignet[2], Joseph B. Leader[3], Danny S. Park[2], Rouel Lanche[1], Evan Maxwell [1], Spencer C. Knight[2], Xiaodong Bai[1], Harendra Guturu[2], Asher Baltzell[2], Ahna R. Girshick[2], Shannon R. McCurdy[2], Raghavendran Partha[2], Adam J. Mansfield[1], David A. Turissini[2], Miao Zhang[2], Joelle Mbatchou [1], Kyoko Watanabe [1], Anurag Verma[4], Giorgio Sirugo[4], Colorado Center for Precision Medicine*, Mayo Clinic Project Generation*, Regeneron Genetics Center*, University of California Los Angeles ATLAS Collaboration*, Marylyn D. Ritchie [4], William J. Salerno[1], Alan R. Shuldiner [1], Daniel J. Rader [4], Tooraj Mirshahi[3], Jonathan Marchini [1], John D. Overton[1], David J. Carey[3], Lukas Habegger[1], Jeffrey G. Reid [1], Aris Economides [1], Christos Kyratsous [1], Katia Karalis[1], Alina Baum[1], Michael N. Cantor [1], Kristin A. Rand[2], Eurie L. Hong[2], Catherine A. Ball[2], Katherine Siminovitch[1], Aris Baras [1,8], Goncalo R. Abecasis [1,8] ✉ & Manuel A. R. Ferreira [1,8] ✉

[1]Regeneron Genetics Center, Tarrytown, NY, USA. [2]AncestryDNA, Lehi, UT, USA. [3]Geisinger, Danville, PA, USA. [4]Department of Genetics, Perelman School of Medicine, University of Pennsylvania, Philadelphia, PA, USA. [8]These authors jointly supervised this work: Aris Baras, Goncalo R. Abecasis, Manuel A. R. Ferreira. *Lists of authors and their affiliations appear at the end of the paper. ✉e-mail: goncalo.abecasis@regeneron.com; manuel.ferreira@regeneron.com

## Colorado Center for Precision Medicine

**Kristy Crooks[5]**

[5]Colorado Center for Personalized Medicine, Aurora, CO, USA. Full lists of members and their affiliations appear in the Supplementary Information.

## Mayo Clinic Project Generation

**James R. Cerhan[6]**

[6]Mayo Clinic, Rochester, MN, USA. Full lists of members and their affiliations appear in the Supplementary Information.

## Regeneron Genetics Center

Jack A. Kosmicki[1], Anthony Marcketta[1], Deepika Sharma[1], Silvio Alessandro Di Gioia[1], Samantha Batista[1], Xiao-Man Yang[1], Gannie Tzoneva[1], Hector Martinez[1], Carlo Sidore[1], Michael D. Kessler[1], Julie E. Horowitz[1], Nilanjana Banerjee[1], Rouel Lanche[1], Evan Maxwell[1], Xiaodong Bai[1], Adam J. Mansfield[1], Joelle Mbatchou[1], Kyoko Watanabe[1], William J. Salerno[1], Alan R. Shuldiner[1], Jonathan Marchini[1], John D. Overton[1], Lukas Habegger[1], Jeffrey G. Reid[1], Aris Economides[1], Christos Kyratsous[1], Katia Karalis[1], Alina Baum[1], Michael N. Cantor[1], Katherine Siminovitch[1], Aris Baras[1,8], Goncalo R. Abecasis[1,8] & Manuel A. R. Ferreira[1,8]

Full lists of members and their affiliations appear in the Supplementary Information.

## University of California Los Angeles ATLAS Collaboration

**Daniel H. Geschwind[7]**

[7]Institute of Precision Health, University of California, Los Angeles, Los Angeles, CA, USA. Full lists of members and their affiliations appear in the Supplementary Information.

## Methods

### Ethics statement

**UKB study.** Ethical approval for the UKB study was obtained from the North West Centre for Research Ethics Committee (no. 11/NW/0382). The work described in this article was approved by the UKB (under application no. 26041).

**GHS study.** The GHS institutional review board (IRB) (no. 2006-0258) approved the DiscovEHR analyses.

**AncestryDNA study.** All data for this research project was from participants who provided previous informed consent to participate in AncestryDNA's Human Diversity Project, as reviewed and approved by their external IRB, Advarra.

**Penn Medicine BioBank study.** Informed consent was obtained from each participant regarding the storage of biological specimens, genetic sequencing and genotyping, and access to all available electronic health record (EHR) data. This study was approved by the University of Pennsylvania IRB and complied with the principles set out in the Declaration of Helsinki (2013).

**Mayo Clinic.** Ethical approval and consent was reviewed and approved by the Mayo Clinic IRB (no. 09-007763).

**Mayo-Regeneron Genetics Center project.** All participants provided informed consent for use of specimens and data in genetic and health research and ethical approval for project generation was provided by the Mayo Clinic IRB (no. 09-007763).

**Colorado Center for Personalized Medicine biobank.** Ethical approval and consent was reviewed and approved by the Colorado Multiple IRB (no. 15-0461).

**University of California Los Angeles.** Patient recruitment and sample collection for precision health activities at University of California Los Angeles (UCLA) was approved by the UCLA IRB (no. 17-001013). Informed consent was obtained for all study participants.

### Phenotype definitions, array genotyping and imputation

**AncestryDNA COVID-19 research study.** US-based AncestryDNA customers over the age of 18 who had consented to the research, were invited to complete five surveys assessing COVID-19 outcomes, as well as providing other demographic information and comorbidities, as described previously[8]. Surveys were released to customers on the following dates: April 2020, June 2020, July 2020, December 2020 and February 2021. About 900,000 customers completed at least one survey (66.4% female, median age 57). Regeneron selected a subset of 300,000 respondents based primarily on COVID-19 and influenza status for inclusion in the GWAS. The specific criteria used to select participants for inclusion and to determine their COVID-19 and influenza status are described in Supplementary Table 10. Briefly, we selected participants who reported: (1) having a positive swab or serology test for SARS-CoV-2; (2) being a first-degree relative of an individual with COVID-19; (3) having a negative swab test for SARS-CoV-2; (4) having a positive flu test in the 2019–2020 or 2020–2021 flu seasons; and (5) survey respondents with no test results for SARS-CoV-2 matched 1:2 or 1:1 to individuals from group (1) based on age, sex, ethnicity and the array type used for genotyping. This ascertainment strategy maximized the number of cases with COVID-19 and matched controls available for analysis.

We selected two survey questions to determine the influenza case status in AncestryDNA: (1) 'The 2019–2020 flu season spans from fall 2019 to late spring 2020. Have you had a flu test in the 2019–2020 flu season?' and (2) 'The 2020–2021 flu season spans from fall 2020 to late spring 2021. Have you had a flu test in the 2020–2021 flu season?' Individuals who responded with 'Yes, and I tested positive' were included as cases in our analysis (Supplementary Table 10). Individuals who responded with 'Yes, and I tested negative' were included as controls for all influenza analyses (initial discovery and the sensitivity analysis, restricting solely to individuals who self-reported an influenza test). Individuals who responded with 'No' were also included as controls for the main discovery influenza analysis.

DNA samples for the 300,000 respondents were genotyped on an Illumina array containing 730,000 SNPs. We removed individuals with discordant sex (based on reported and genetically determined sex) and those with <98% sample call rate[8]. We removed array variants with allele frequency differences greater than 0.1 between array versions, as well as variants with a call rate lower than 98%. Variants were then imputed with the Haplotype Reference Consortium reference panel (v.1.1). We determined best-guess haplotypes with Eagle (v.2.4.1) and performed imputation with Minimac4 (v.1.0.1). From 11,117,080 variants, we retained 8,049,082 imputed variants ($r^2 > 0.3$) in the final dataset.

**GHS DiscovEHR study.** The GHS MyCode Community Health Initiative is a health system-based cohort from Pennsylvania with ongoing recruitment since 2006. Participants were genotyped on either the Illumina OmniExpress Exome (OMNI) or Global Screening Array (GSA) and imputed to the TOPMed reference panel (stratified according to array) using the TOPMed Imputation Server. Before imputation, we retained variants with a minor allele frequency (MAF) ≥ 0.1%, missingness < 1% and Hardy–Weinberg (HWE) $P > 10^{-15}$. After imputation, data from the OMNI and GSA datasets were merged for subsequent association analyses, which included an OMNI and GSA batch covariate, in addition to the other covariates described below. ICD-10-based influenza case status was defined using a combination of the following three two-digit ICD-10 codes and their nested three-digit and four-digit codes: J09 (influenza due to certain identified influenza viruses), J10 (influenza due to other identified influenza viruses) and J11 (influenza due to unidentified influenza viruses) (Supplementary Table 11). Using these ICD-10 codes, we defined as influenza cases individuals with (1) one or more inpatient record of influenza or (2) two or more outpatient records of influenza. Influenza controls were all other individuals with available genotype data, except for 722 individuals not identified as cases but who had a positive cell culture assay for influenza A or B, as described below; these individuals were excluded from the analysis.

A subset of 82,348 individuals had viral cell culture assays that included influenza A and B. Of these, 1,694 and 235 individuals had a positive assay for influenza A and B, respectively. Lastly, we used inpatient hospital records to identify 528 individuals with influenza listed as the primary cause of hospitalization, using the ICD-10 codes listed above (Supplemental Table 11).

**UKB study.** The UKB study includes approximately 500,000 adults aged 40–69 at recruitment between 2006 and 2010. DNA samples were genotyped using the Applied Biosystems UK BiLEVE Axiom array ($n = 49,950$) or the Applied Biosystems UK Biobank Axiom array ($n = 438,427$). Genotype data for variants not included in the arrays were inferred using the TOPMed reference panel, as described above. Influenza case status was defined in the same way as with all the other ICD-10-based biobanks (see Supplementary Table 11 for a full list of ICD-10 codes and case sample sizes). As with the GHS, we used inpatient hospital records to identify 1,168 individuals with influenza listed as the primary cause of hospitalization, using the ICD-10 codes listed above (Supplemental Table 11).

**Penn Medicine BioBank study, Colorado Center for Personalized Medicine biobank, Mayo Clinic biobank and University of California Los Angeles ATLAS Precision Health Biobank.** The Penn Medicine BioBank (PMBB) contains approximately 70,000 study participants,

all recruited through the University of Pennsylvania Health System. Participants donate blood or tissue and allow access to EHR information. The Colorado Center for Personalized Medicine (CCPM) biobank at the University of Colorado Anschutz Medical Campus in Aurora encompasses approximately 45,000 individuals (aged 30–92). De-identified phenotype data were collected by Health Data Compass and include an individual's entire medical record from the University of Colorado's EHR. Project generation included 116,277 individuals from the Mayo Clinic biobank (ongoing enrollment from 2009) and 30 disease-specific registries. Structured data in the EHR were extracted using the Observational Medical Outcomes Partnership common data model (https://www.ohdsi.org/data-standardization/). The ATLAS Precision Health Biobank at UCLA comprises approximately 32,000 individuals. De-identified phenotype data include all hospital visits beginning in 2013 and converted to ICD-10 codes. Influenza case status for these biobanks was defined in the same way as described above (see Supplementary Table 11 for a full list of ICD-10 codes and case sample sizes).

## Exome sequencing

**Sample preparation and sequencing.** Exome capture was completed at the Regeneron Genetics Center. Briefly, samples were pooled before exome capture, with either (1) a slightly modified version of the xGen probe library (Integrated DNA Technologies (IDT); UKB, PMBB and 81,620 samples of the DiscovEHR), (2) NimbleGen VCRome (58,856 samples of DiscovEHR) or (3) the TWIST human compressive exome panel for CCPM, Mayo Clinic and UCLA. The multiplexed samples were sequenced using: (1) for the UKB samples—75-bp paired-end reads with two 10-bp index reads on the Illumina NovaSeq 6000 platform using S2 or S4 flow cells; (2) for the DiscovEHR samples captured with VCRome—75-bp paired-end reads with two 8-bp index reads on the Illumina HiSeq 2500 platform; (3) for the DiscovEHR captured with the IDT system—two 8-bp index reads on the Illumina HiSeq 2500 platform or two 10-bp index reads on the Illumina NovaSeq 6000 platform on S4 flow cells; or (4) for the PMBB, CCPM, Mayo Clinic and UCLA—two 10-bp index reads on the Illumina NovaSeq 6000 platform on S4 flow cells.

**Variant calling and quality control.** Sample read mapping, variant calling, aggregation and quality control were performed using Deep-Variant. Briefly, NovaSeq whole-exome sequencing reads were mapped with the Burrows–Wheeler Aligner MEM to the hg38 reference genome. DeepVariant identified small variants and reported them as per-sample genomic variant call format (VCF); they were aggregated into a jointly genotyped, multisample VCF. After aggregation of genotypes, we trained a support vector machine model on several summary-level per-site metrics to distinguish poor from higher-quality variants.

**Gene burden tests.** Briefly, for each gene region as defined by Ensembl[22], genotype information from multiple rare coding variants was collapsed into a single burden genotype, such that individuals who were: (1) homozygous reference for all variants in that gene were considered homozygous reference; (2) heterozygous for at least one variant in that gene were considered heterozygous; and (3) only individuals that carried two copies of the alternative allele of the same variant were considered homozygous for the alternative allele. We did this separately for seven classes of variants: (1) predicted LOF (frameshift, splice acceptor and donor, and stop gained variants); (2) predicted LOF or missense; (3) predicted LOF or missense variants predicted to be deleterious by at least 1 of 5 algorithms; (4) predicted LOF or missense variants predicted to be deleterious by 5 of 5 algorithms; (5) missense; (6) missense variants predicted to be deleterious by 1 of 5 algorithms; (7) missense variants predicted to be deleterious by 5 of 5 algorithms. Variants were annotated using VEP and the canonical transcript. The five missense deleterious algorithms used were SIFT[23], PolyPhen-2 (HDIV), PolyPhen-2 (HVAR)[24], LRT[25] and MutationTaster[26]. For each

gene, and for each of these seven groups, we considered five separate burden masks based on the alternative allele frequency of the variants collapsed into the burden genotype: <1%, <0.1%, <0.01%, <0.001% and singletons only. Each burden mask was tested for association with the same approach used for the individual variants.

## Genetic association analyses

Association analyses were performed using the REGENIE[9] genome-wide Firth logistic regression test. We included in step 1 of REGENIE (that is, prediction of individual trait values based on the genetic data) directly genotyped (imputed for the GHS) variants with an MAF > 1%, <10% missingness, HWE $P > 10^{-15}$ and LD pruned (1,000 variant windows, 100 variant sliding windows and $r^2 < 0.9$). The association model used in step 2 of REGENIE included the covariates of age, $age^2$, sex, age × sex, $age^2 × sex$ and the first ten principal components (PCs) derived from the analysis of a stricter set of LD-pruned (1,000 variant windows, 50 variant step size and $r^2 < 0.9$) common variants from the array (imputed for the GHS) data. For both individual rare variants and burden masks, we used the same covariates as in the GWAS but added 20 PCs from rare variants[27–29] and (when appropriate) sequencing batch covariates.

Within each study, association analyses were performed separately for five ancestral groups defined based on genetic similarity with samples from the five superpopulations studied by the 1000 Genomes Project: from Africa (AFR), the Americas (AMR), East Asia (EAS), Europe (EUR) and South Asia (SAS). As such, these five subgroups can be thought of as 1000 Genomes-like ancestral superpopulations. Genetic similarity was defined by projecting each sample onto reference PCs calculated from the HapMap3 reference panel. Briefly, we merged our samples with HapMap3 samples and kept only SNPs in common between the two datasets. We excluded SNPs with an MAF < 10%, genotype missingness greater than 5% or HWE $P < 10^{-5}$. We calculated PCs for the HapMap3 samples and projected each sample onto those PCs. To assign a group to each non-HapMap3 sample, we trained a kernel density estimator using the HapMap3 PCs and used the kernel density estimators to calculate the likelihood of a given sample belonging to each of the five groups. When the likelihood for a given group was greater than 0.3, we assigned the sample to that group. When a sample had two group likelihoods greater than 0.3, we arbitrarily assigned 1000 Genomes-like AFR over 1000 Genomes-like EUR ($n_{AncestryDNA} = 0$; $n_{CCPM} = 0$; $n_{GHS} = 36$; $n_{Mayo Clinic} = 0$; $n_{UCLA} = 0$; $n_{UKB} = 56$; $n_{PMBB} = 7$), 1000 Genomes-like AMR over 1000 Genomes-like EUR ($n_{AncestryDNA} = 1,953$; $n_{CCPM} = 489$; $n_{GHS} = 455$; $n_{Mayo Clinic} = 358$; $n_{UCLA} = 497$; $n_{UKB} = 436$; $n_{PMBB} = 138$), 1000 Genomes-like AMR over 1000 Genomes-like EAS ($n_{AncestryDNA} = 0$; $n_{CCPM} = 0$; $n_{GHS} = 2$; $n_{Mayo Clinic} = 0$; $n_{UCLA} = 0$; $n_{UKB} = 2$; $n_{PMBB} = 1$), 1000 Genomes-like SAS over 1000 Genomes-like EUR ($n_{AncestryDNA} = 617$; $n_{CCPM} = 24$; $n_{GHS} = 32$; $n_{Mayo Clinic} = 34$; $n_{UCLA} = 89$; $n_{UKB} = 592$; $n_{PMBB} = 36$) and 1000 Genomes-like AMR over 1000 Genomes-like AFR ($n_{AncestryDNA} = 5$; $n_{CCPM} = 3$; $n_{GHS} = 192$; $n_{Mayo Clinic} = 0$; $n_{UCLA} = 4$; $n_{UKB} = 51$; $n_{PMBB} = 77$). We excluded samples from the analysis if no genetic ancestry likelihoods were greater than 0.3, or if more than three genetic ancestry likelihoods were >0.3 ($n_{AncestryDNA} = 2,947$; $n_{CCPM} = 774$; $n_{GHS} = 821$; $n_{Mayo Clinic} = 391$; $n_{UCLA} = 837$; $n_{UKB} = 1,205$; $n_{PMBB} = 384$).

We performed an inverse-variance-weighted meta-analysis to combine association results across genetic ancestries and studies and used Cochran's $Q$ to assess the heterogeneity of effect sizes between contributing studies and genetic ancestries. Within the text, we reported $P$ values from the Cochran's $Q$ test as unadjusted heterogeneity $P$ values.

## LD score regression

LD score regression[12] was used to estimate genetic correlations[30] between influenza and summary statistics of two COVID-19 phenotypes from the HGI[2], that is, SARS-CoV-2 infection (C2) and COVID-19 hospitalization (B2). As LD score regression depends on matching the LD structure of the analysis sample to a reference panel, we used the phenotypes and corresponding summary statistics available in

Europeans by the HGI[2], which is why severe COVID-19 (A2) was not used. We conducted analyses using the standard program settings for variant filtering (removal of non-HapMap3 SNPs, non-autosomal, chi-squared > 30, MAF < 1% or allele mismatch with reference). Differences between the observed genetic correlations were compared using z-scores.

### Impact of *ST6GAL1* and *B3GALT5* knockdown on influenza infectivity in vitro

**Cell culture.** A549 and Calu-3 cells were purchased from ATCC and maintained in F-12 medium supplemented with 10% heat-inactivated FCS, and MEM containing Earle's Balanced Salts, L-glutamine nonessential amino acids, sodium pyruvate and 10% FCS, respectively. Cells were tested periodically for mycobacterial contamination.

**siRNA knockdown.** Lipofectamine RNAiMAX and four Silencer Select siRNAs at a final concentration of 10 μM were used for the knockdown experiments according to the manufacturer's protocol. siRNA1 no. s12841 and siRNA2 no. s12843 were used to knock down *ST6GAL1*, while siRNA1 no. s20171 and siRNA2 no. s20172 were used to knock down *B3GALT5*. A nontargeting Cy3-conjugated siRNA (control no. 1) was used as the negative control and to optimize the transfection conditions. siRNA targeting *GAPDH* (cat. no. 4390849, Thermo Fisher Scientific) was used to control the specificity of the effect. On the day of the experiments, cells were seeded in 12-well microplates and transfected in triplicate for each siRNA and control. Forty-eight hours after transfection cells were detached with TrypLE, wells were combined; cells were counted for seeding and plated at 20,000 cells per well in black, clear-bottom 96-well plates, with at least eight replicates for each siRNA condition, and incubated at 37 °C overnight for the infection experiment. The other cells were then seeded in a well of a 12-well microplate and incubated at 37 °C overnight to test for knockdown efficiency.

**In vitro influenza infection.** Seventy-two hours after transfection, H1N1 influenza virus A (Puerto Rico/8/34) expressing green fluorescent protein (GFP) (PR8-GFP) was thawed on ice. In infection medium (DMEM containing 3% FCS and 10% penicillin-streptomycin glutamine), PR8-GFP was diluted to a concentration representing a multiplicity of infection (MOI) of 10. The virus was then serially diluted 1:3 to a final MOI of 0.01. Knockdown cells were then removed from the incubator and the medium was removed from the cells. Then, 100 μl of diluted virus or medium alone was added to the wells; then cells were further incubated at 37 °C for 18–24 h. After that time, virus-containing medium was removed from the cells and each well was overlayed with 100 μl 1× PBS. Plates were imaged on a SpectraMax i3 with MiniMax to measure infection by quantifying GFP+ cells. Percentage infection was then calculated and normalized to the negative control cells at each MOI as 100%. Data were graphed using Prism v.9.3 (GraphPad Software).

**RNA extraction and quantitative PCR.** Seventy-two hours after transfection, RNA was extracted using RNeasy PLUS Mini kit. Complementary DNA was synthetized using the SuperScript IV VILO Master Mix and the knockdown levels were evaluated using QuantStudio 6 PCR system with specific TaqMan probes (*ST6GAL1* assay ID Hs00949382_m1; *ACTB* assay ID Hs01060665_g1; *GAPDH* assay ID Hs02786624_g1; assay ID *B3GALT5* Hs00707757_s1). Data were analyzed with the Analysis Software v.2.6 for QuantStudio 6. Plots and statistics were generated with Prism v.9.3.

**Membrane sialic acid staining.** siRNA-transfected cells and controls were dissociated 72 h after transfection using TrypLE and incubated for 15 min at room temperature in the dark with fluorescein-conjugated *Sambucus Nigra* Lectin at a final concentration of 2 μg ml⁻¹. After two washes with PBS, membrane fluorescence was evaluated using the

CytoFLEX LX cytometer. Raw cytofluorimeter data were analyzed using FlowJo v.10.8.0; graphs, representing the mean intensity of three wells, were generated using Prism v.9.3.

**Immunoblot.** Total lysate from siRNA-treated A549 cells was extracted using radioimmunoprecipitation assay buffer (cat. no. 89900) and quantified using the Bio-Rad Laboratories DC protein assay (cat. no. 5000111). For each sample, 25 μg of total lysate were loaded on a well of 4–12% NuPAGE Bis-Tris gel (cat. no. NP0323BOX) and run using MES running buffer (cat. no. NP0002). After blotting, membranes were blocked in blocking buffer (5% milk plus 3% BSA in tris-buffered saline with Tween 20) for 1 h at room temperature and goat anti-ST6GAL1 antibody (cat. no. AF5924, R&D Systems) diluted 1:200 in blocking buffer at 4 °C overnight. Secondary chicken anti-goat horseradish peroxidase (HRP)-conjugated antibody was used for blotting (cat. no. HAF019, R&D Systems) diluted 1:1,000 in blocking buffer. β-Actin HRP (diluted 1:10,000, 30 min, cat. no. 5123, Cell Signaling Technology) or GAPDH HRP (1:10,000, 30 min, cat. no. HRP-60004, Proteintech) were used as the loading control. The chemiluminescence signal was detected using a ChemiDoc imager (Bio-Rad Laboratories).

### Reporting summary

Further information on research design is available in the Nature Portfolio Reporting Summary linked to this article.

### Data availability

Summary statistics are available via the GWAS Catalog (accession no. GCST90432107). Individual-level exome sequencing, genotype and phenotype data are available to approved researchers via the UKB at https://www.ukbiobank.ac.uk/enable-your-research. The FinnGen release 8 influenza GWAS summary statistics are available to approved researchers at https://www.finngen.fi/en/access_results. The influenza A seropositivity GWAS summary statistics were downloaded from the GWAS Catalog (accession no. GCST006339). Precalculated LD scores from the 1000 Genomes[10] European reference population were obtained from https://data.broadinstitute.org/alkesgroup/LDSCORE/. GTEx data can be accessed at https://gtexportal.org/. Source data are provided with this paper.

### Code availability

Genetic data are represented in the PLINK format (v.1.90b6.21), which is available at https://www.cog-genomics.org/plink2/, and were analyzed using REGENIE (v.3.1.3), which is available at https://github.com/rgcgithub/regenie. Meta-analyses were performed using METAL (v.2020-05-05), which is available at https://github.com/statgen/METAL. Imputation was done with Minimac4 (v.1.01), which is available at https://github.com/statgen/Minimac4. Read alignment was performed using the Burrows–Wheeler Aligner (v.0.7.17) available at http://bio-bwa.sourceforge.net. Picard (v.1.141) was used for duplicate marking and is available at https://broadinstitute.github.io/picard/. SAM, BAM and CRAM file generation and manipulation was performed using Samtools (v.1.7), which is available at http://www.htslib.org. Variant calling was performed using weCall (v.1.1.2), which is available at https://github.com/Genomicsplc/wecall. VCF file manipulation and index generation was performed using BCFtools (v.1.7), which is available at http://www.htslib.org. All other data analyses were performed using Python (v.3.8), R (v.4.0.4), Prism v.9.3.0 and QuantStudio 6 (v.2.6). R packages used include ggplot2 (v.3.4.2) and patchwork (v.1.1.3). Python packages used include pandas (v.2.0.3) and numpy (v.1.25.2).

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

## Acknowledgements

This research was conducted using the UKB resource (project no. 26041). The UCLA ATLAS Collaboration is supported by the David Geffen School of Medicine, UCLA Health, and the UCLA Clinical and Translational Science Institute (no. UL1TR001881). The PMBB is funded by a gift from the Smilow family, the National Center for Advancing Translational Sciences of the National Institutes of Health under Clinical and Translational Science Award no. UL1TR001878, and the Perelman School of Medicine at the University of Pennsylvania. We thank the participants and investigators of the FinnGen study. We also thank the AncestryDNA customers who voluntarily contributed information in the COVID-19 survey.

## Author contributions

J.A.K. performed the GWAS, meta-analysis and statistical analyses. S.A.D.G. and S.B. performed the experimental analyses. S.A.D.G., S.B., X.-M.Y., H.M. and C.K. provided the materials and reagents. J.A.K., D.S., N.B. and M.N.C. created the phenotypes. J.A.K. and M.A.R.F. wrote the manuscript and created the figures. G.T., M.D.K., A.R.S., C.K., K.K., A. Baum and J.E.H. assisted with the biological interpretation. C.S. performed the eQTL analysis. J. Mbatchou, K.W. and J. Marchini assisted with the statistical analyses. A.E., K.S., A. Baras, G.R.A. and M.A.R.F. supervised the study. G.H.L.R., M.V.C., D.S.P., S.C.K., H.G., A. Baltzell, A.R.G., S.R.M., R.P., D.A.T., M.Z., K.A.R., E.L.H. and C.A.B. provided the data from AncestryDNA. A.E.J., J.B.L., D.J.C. and T.M. provided the data from the GHS. A.V., G.S., M.D.R. and D.J.R. provided the data from the PMBB. A.M., R.L., E.M., A.J.M., X.B., W.J.S., J.D.O., L.H. and J.G.R. performed DNA extraction, genotyping, imputation and variant calling. All authors reviewed and approved the final version of the manuscript.

## Competing interests

J.A.K., A.M., D.S., S.A.D.G., S.B., X.-M.Y., G.T., H.M., C.S., M.D.K., J.E.H., N.B., R.L., E.M., X.B., A.J.M., J. Mbatchou, K.W., W.J.S., A.R.S., J. Marchini, J.D.O., L.H., J.G.R., A.E., C.K., K.K., A. Baum, M.N.C., K.S., A. Baras, G.R.A. and M.A.R.F. are current employees or stockholders of Regeneron Genetics Center or Regeneron Pharmaceuticals. G.H.L.R., M.V.C., D.S.P., S.C.K., H.G., A. Baltzell, A.R.G., S.R.M., R.P., D.A.T., M.Z., K.A.R., E.L.H. and C.A.B. are current or past employees of AncestryDNA and may hold equity in AncestryDNA. The other authors declare no competing interests.

## Additional information

**Extended data** is available for this paper at https://doi.org/10.1038/s41588-024-01844-1.

**Correspondence and requests for materials** should be addressed to Goncalo R. Abecasis or Manuel A. R. Ferreira.

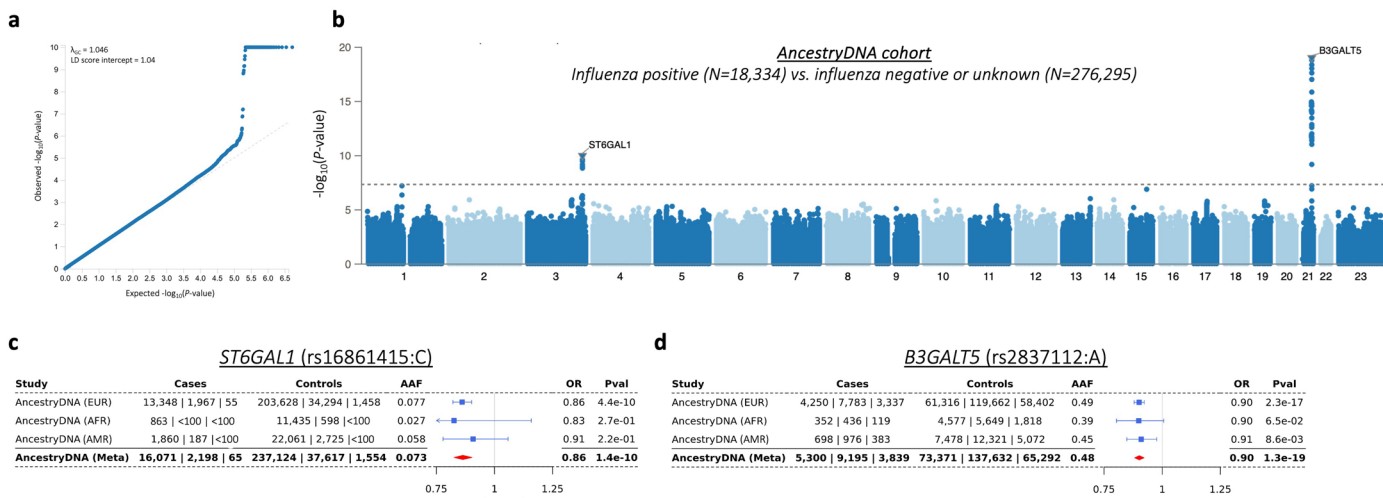

**Extended Data Fig. 1 | Results from the discovery GWAS of reported influenza infection in the AncestryDNA cohort.** We tested 10 million common (alternate allele frequency [AAF] > 1%) variants, derived from array genotyping followed by HRC imputation, comparing 18,334 individuals who reported a positive test for influenza (cases) against 276,295 individuals who did not report a positive test for influenza (controls). **a**, Quantile-quantile plot showing observed *P*-values for individual variants (*y*-axis) against *P*-values expected by chance given multiple testing (*x*-axis). The genomic inflation factor ($\lambda_{GC}$) of this analysis was 1.05,

whereas the intercept from LD-score regression was 1.04. **b**, Manhattan plot showing association ($-\log_{10}$ *P*-value) with imputed variants. The dotted grey line demarcates the genome-wide significance threshold of $P = 5 \times 10^{-8}$. **c,d**, Genetic ancestry-specific results for the 3q27.3/*ST6GAL1* (**c**) and the 21q22.2/*B3GALT5* (**d**) variants. Unadjusted *P*-values derived from Firth-regression (two-sided test) implemented in REGENIE[9]. Error bars represent the 95% confidence interval around the odds ratio (data point).

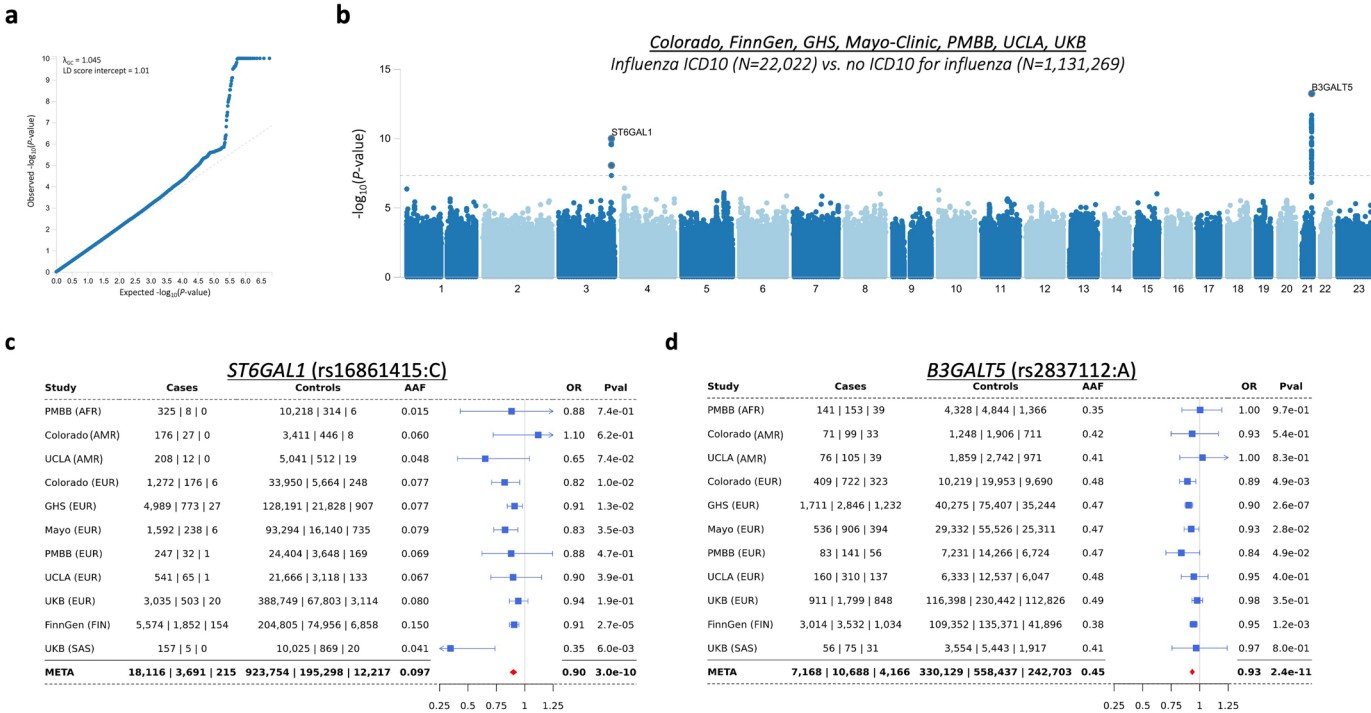

**Extended Data Fig. 2 | Summary of results from the replication GWAS of lifetime medical record-based influenza infection performed across seven biobanks.** We tested 11 million common (AAF > 1%) variants derived from array genotyping followed by TOPMed imputation (except FinnGen, which used an imputation reference panel comprising samples from Finland), comparing 22,022 individuals with (cases) against 1,131,269 individuals without (controls) an ICD-10 code for influenza (controls). **a**, Quantile-quantile plot showing observed *P*-values for individual variants (*y*-axis) against *P*-values expected by chance given multiple testing (*x*-axis). The genomic inflation factor ($\lambda_{GC}$) of this analysis was 1.04, whereas the intercept from LD-score regression was 1.01. **b**, Manhattan plot showing association (−log$_{10}$ *P*-value) with imputed variants. The dotted grey line demarcates the genome-wide significance threshold of $P = 5 \times 10^{-8}$. **c**,**d**, Cohort-specific results for the 3q27.3/*ST6GAL1* (**c**) and the 21q22.2/*B3GALT5* (**d**) variants. Unadjusted *P*-values derived from Firth-regression (two-sided test) implemented in REGENIE[9]. Error bars represent the 95% confidence interval around the odds ratio (data point).

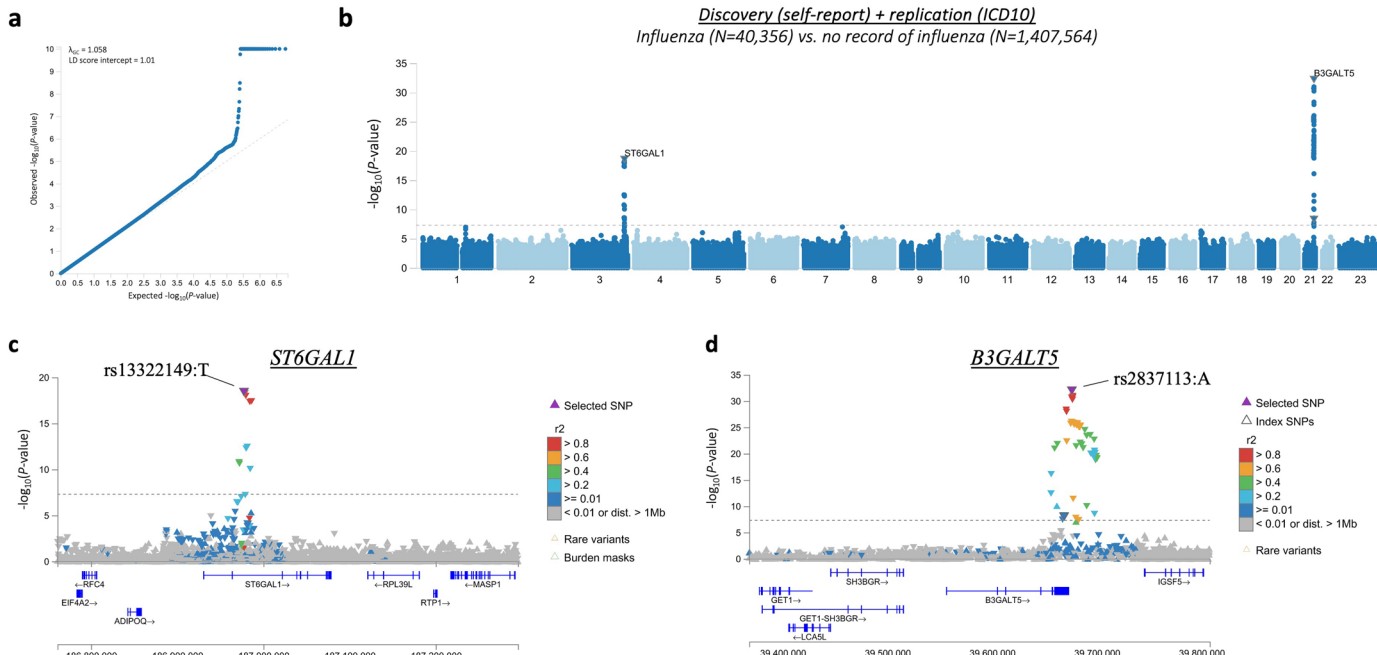

**Extended Data Fig. 3 | Meta-analysis of the discovery (reported positive test in AncestryDNA) and replication (lifetime medical record across seven biobanks) GWAS of influenza. a**, Quantile-quantile plot showing observed *P*-values for individual variants (*y*-axis) against *P*-values expected by chance given multiple testing (*x*-axis). The genomic inflation factor (λ_GC) of this analysis was 1.06, whereas the intercept from LD-score regression was 1.02. **b**, Manhattan plot showing association ( − log₁₀*P*-value) with imputed variants. The dotted grey line demarcates the genome-wide significance threshold of $P = 5 \times 10^{-8}$. **c,d**, Regional associations plots for the 3q27.3/*ST6GAL1* (**c**) and 21q22.2/*B3GALT5* (**d**) loci. Variants are colored based on their linkage disequilibrium ($r^2$) with the lead variant (purple triangle). Upward facing triangles represent variants with OR > 1, and downward facing triangles represent OR < 1. Unadjusted *P*-values derived from Firth-regression (two-sided test) implemented in REGENIE[9].

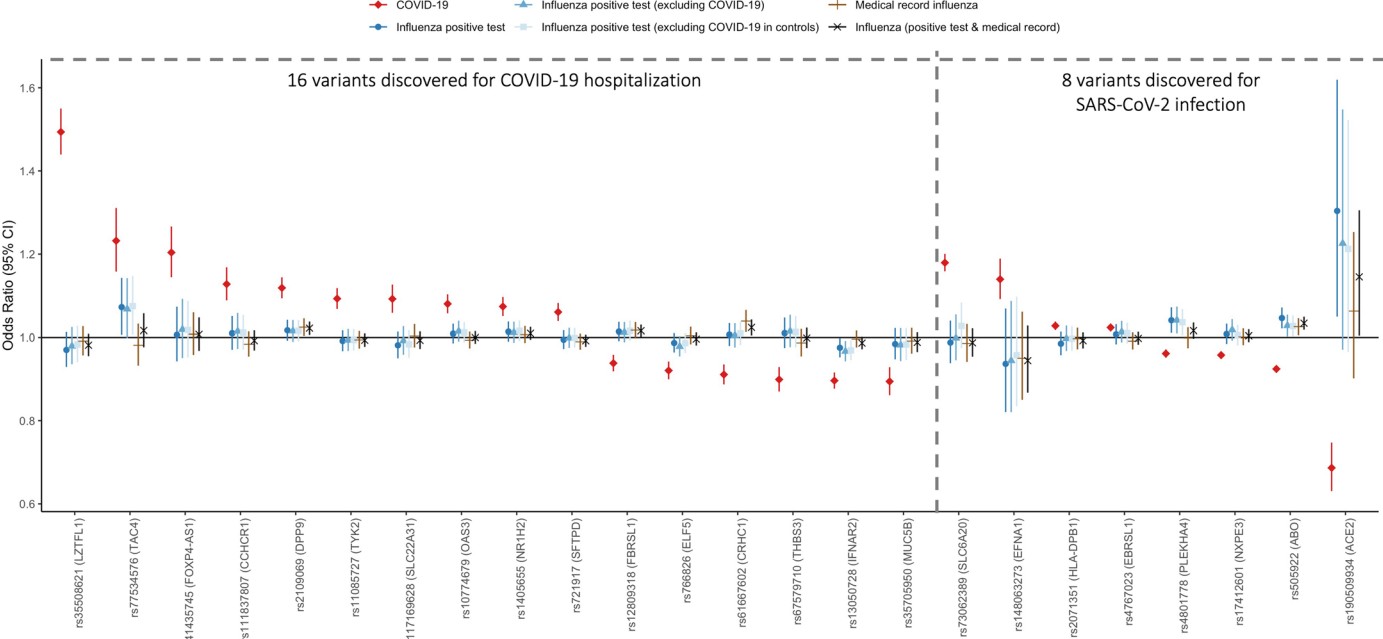

**Extended Data Fig. 4 | Association between COVID-19 risk variants and influenza infection in AncestryDNA (18,334 cases vs. 276,295 controls), biobank cohorts (22,022 cases vs. 1,131,269 controls) or overall meta-analysis (40,356 cases vs. 1,407,564 controls).** Of the 24 COVID-19 risk variants, 16 were discovered in a GWAS of COVID-19 hospitalization (comparing COVID-19 hospitalized cases against individuals with no record of SARS-CoV-2 infection), and 8 were discovered in a GWAS of reported SARS-CoV-2 infection (comparing all individuals with a record of SARS-CoV-2 infection against individuals with no record of SARS-CoV-2 infection). The association observed between the 24 variants and influenza infection was comparable between AncestryDNA, biobank cohorts and overall meta-analysis GWAS. Error bars represent the 95% confidence interval around the odds ratio (data point).

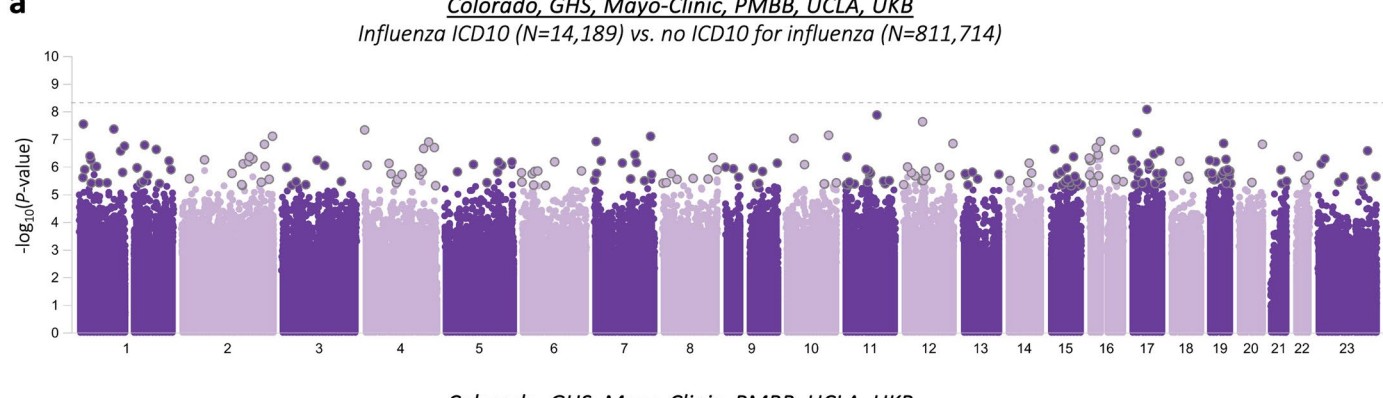

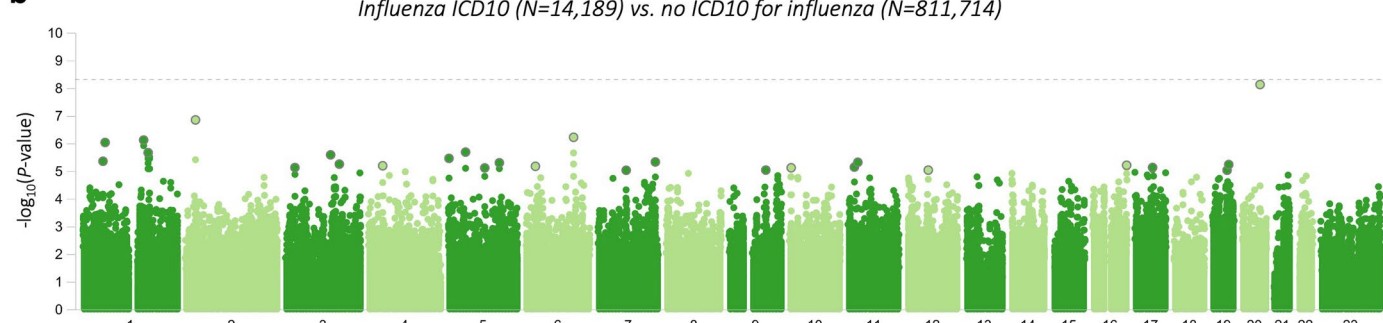

**Extended Data Fig. 5 | Summary of association results between lifetime medical record-based influenza and rare coding variants from exome sequencing in six biobanks (Colorado, DiscovEHR, Mayo-Clinic, UCLA, UKB, UPENN-PMBB).** We tested 23 million rare (AAF < 1%) variants derived from exome sequencing, comparing 14,189 individuals with (cases) against 811,714 individuals without (controls) an ICD10 code for influenza. **a**,**b**, Manhattan plots of (**a**) individual coding variants (each point represents a single variant) and (**b**) coding variants tested on aggregate through gene burden tests (each point represents a burden test for a gene, with up to 40 different burden tests performed per gene; Methods). The dotted grey line demarcates $P = 2.1 \times 10^{-9}$ (corresponding to a Bonferroni correction for the number of individual variant and gene-based burden tests performed). Unadjusted $P$-values derived from Firth-regression (two-sided test) implemented in REGENIE[9].

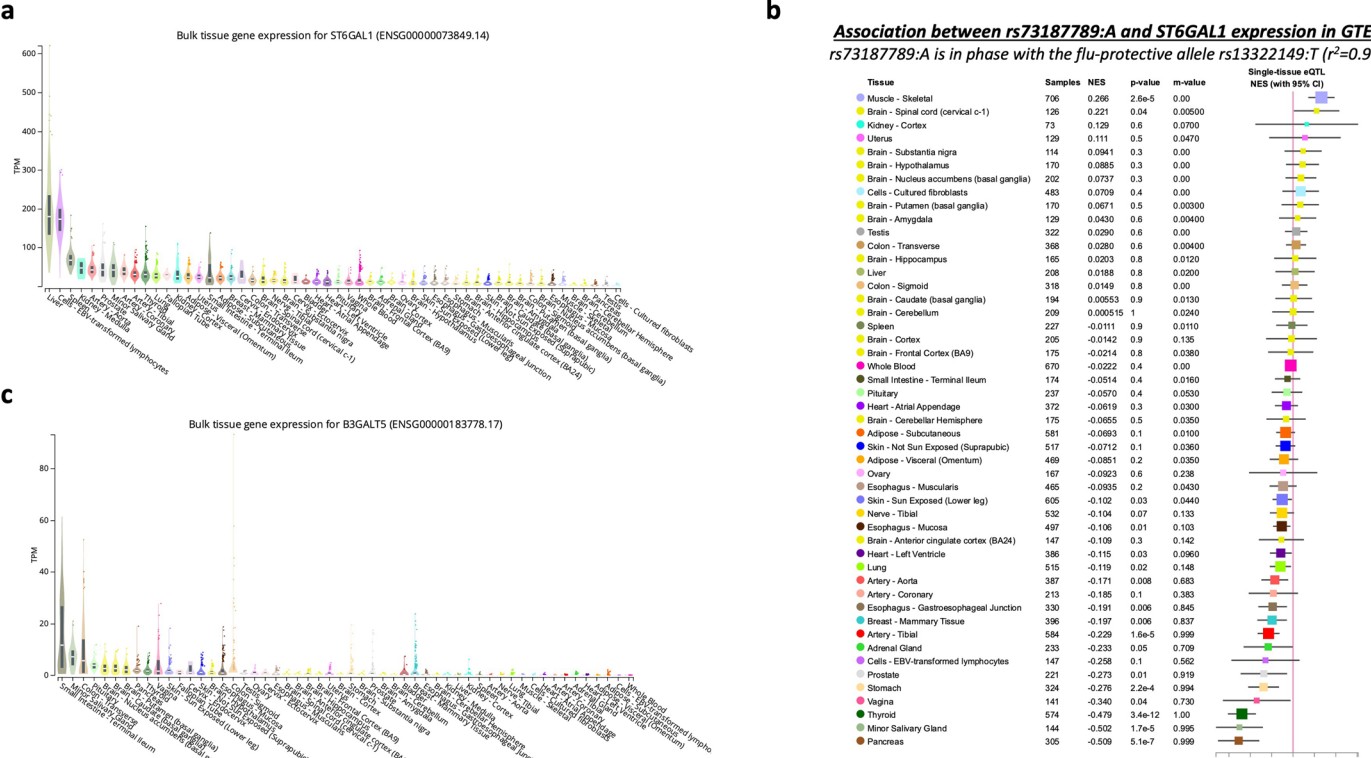

**Extended Data Fig. 6 | Expression of *ST6GAL1* and *B3GALT5* across human tissues measured by the GTEx consortium. a**, Expression levels expressed as transcripts per million (TPM) per tissue in GTEx for *ST6GAL1*. Box plots show the interquartile range (ICR) and the median. Sample sizes for each tissue can be found on the GTEx website (see **Data Availability**). **b**, Association between rs73187789:A and expression levels of *ST6GAL1* across tissues. Variant rs73187789

was a lead independent eQTL for *ST6GAL1* in thyroid tissue and was in high LD ($r^2$ = 0.95) with the lead variant associated with risk of influenza in *ST6GAL1*. Error bars represent the 95% confidence interval around the normalized effect size (NES) from linear regression. **c**, Expression levels expressed as TPM per tissue in GTEx for *B3GALT5*. Box plots show the ICR and the median. Sample sizes for each tissue can be found on the GTEx website (see **Data Availability**).

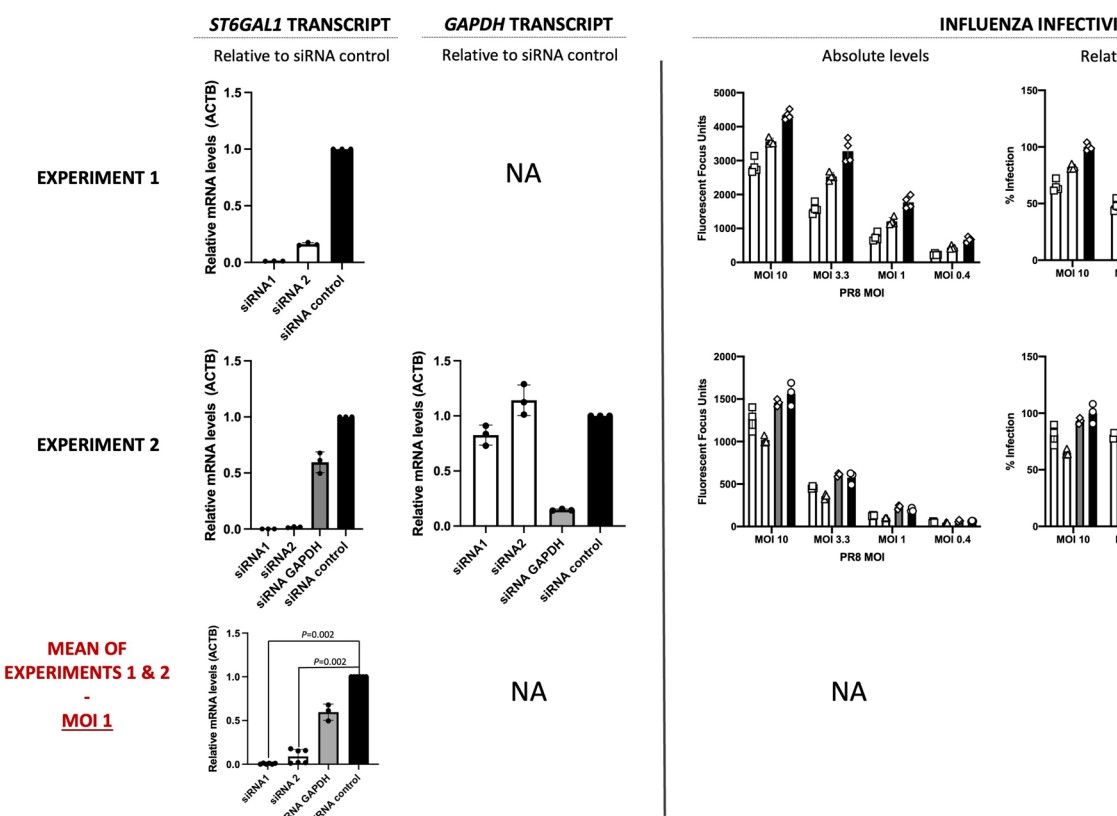

**Extended Data Fig. 7 | Impact of siRNA knockdown of *ST6GAL1* in A549 cells on influenza infectivity.** In each plot, bars represent the averages across replicates from each experiment (*n* = 2), points represent the values from the individual replicates (*n* = 4 for experiment 1; *n* = 3 for experiment 2), and lines show the width of the distribution of the data points. The first column shows mRNA levels of *ST6GAL1* relative to *ACTB* transcript in cells treated with four different siRNAs: two targeting *ST6GAL1* (siRNA1 and siRNA2) and two negative controls (one targeting *GAPDH* [experiment 2 only] and a scrambled siRNA). The second and third columns show results from infection assay with PR8-GFP (H1N1, multiplicity of infection [MOI] of 0.4 to 10), with the latter column showing infectivity relative to the scrambled siRNA control. The *GAPDH* siRNA (but not the two siRNAs against *ST6GAL1*) significantly reduced GAPDH expression relative to the scrambled siRNA (~80% reduction). *P*-values derived from a two-sided Wilcoxon Rank Sum Test and asterisks (*) mark those experiments with *P* < 0.05.

Effect of *ST6GAL1* and control siRNAs on:

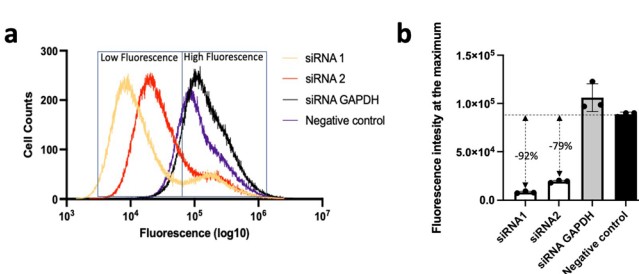

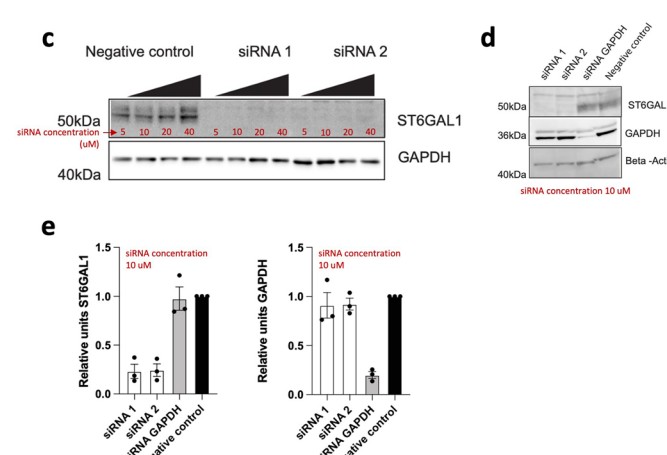

**Extended Data Fig. 8 | Impact of *ST6GAL1* siRNA knockdown on sialic acid abundance. a**, Flow cytometry histograms (geometric mean across three replicates) of siRNA-transfected cells stained with FITC-conjugated *S. Nigra* (SNA) Lectin 72 hours post-transfection to measure membrane-level sialic acid. A small proportion of cells treated with *ST6GAL1* siRNAs displayed high fluorescence levels, consistent with incomplete transfection. **b**, Bar graph showing mean fluorescence intensity at the maximum from histogram in **a**. Bars represent the average across replicates from three experiments, points represent values from the individual replicates, and lines show the width of the distribution of individual experiments. In comparison to the negative control, membrane-level

sialic acid dropped by 79–92% after *ST6GAL1* knockdown. **c,d**, Representative images of ST6GAL1 and GAPDH protein levels measured in A549 cells 72 hours after treatment with siRNAs at concentration 5 to 40 μM (**c**) and at the final selected concentration of 10 μM (**d**). **e**, Quantification of ST6GAL1 and GAPDH protein levels in A549 cells treated with 10 μM of *ST6GAL1* siRNA, based on three individual replicates. Protein levels are normalized to beta-actin and shown relative to the negative control siRNA. Uncropped gels are provided as **Source Data**. *P*-values derived from a two-sided Wilcoxon Rank Sum Test and asterisks (\*) mark those experiments with *P* < 0.05.

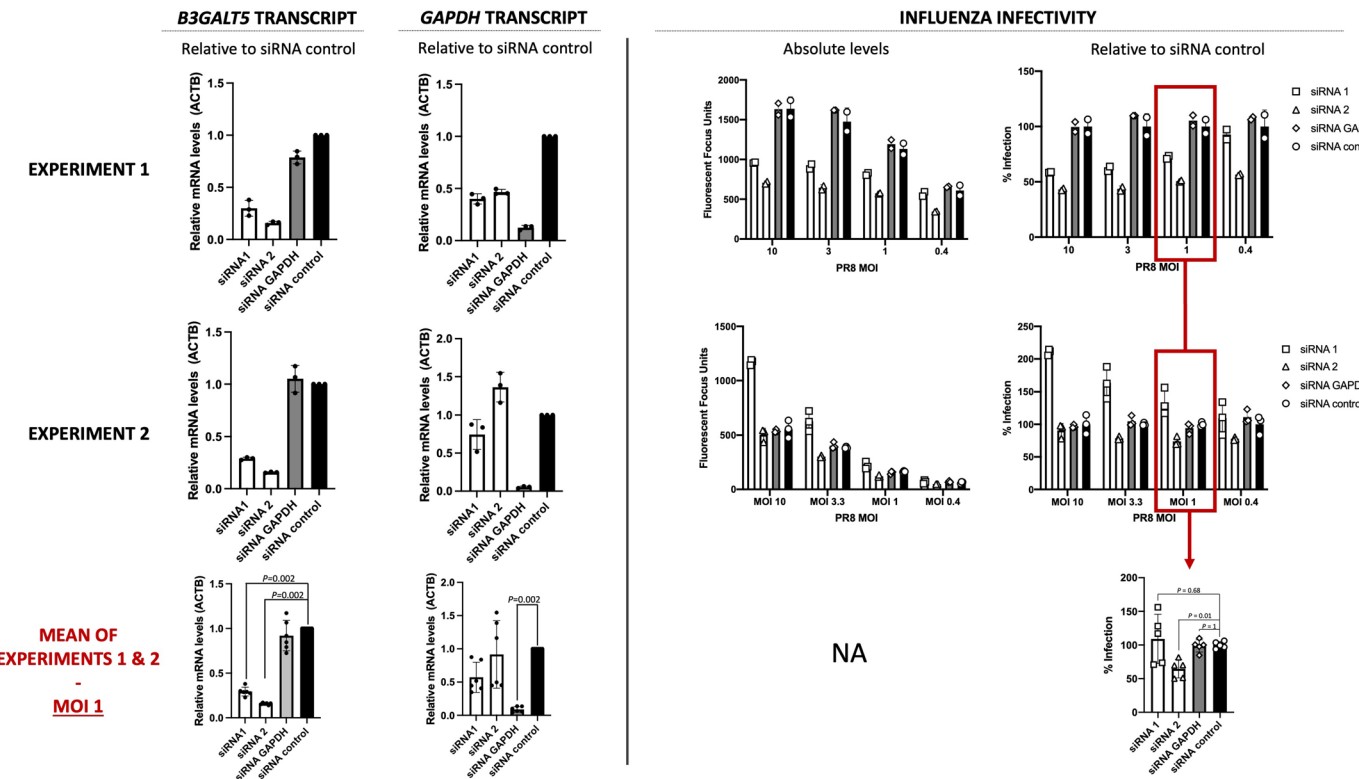

**Extended Data Fig. 9 | Impact of *B3GALT5* siRNA knockdown on influenza infectivity in Calu-3 cells.** In each plot, bars represent the averages across replicates from each experiment (*n* = 2), points represent the values from the individual replicates (*n* = 3), and lines show the width of the distribution of the data points. The first column shows mRNA levels of *B3GALT5* relative to *ACTB* transcript in cells treated with four different siRNAs: two targeting *B3GALT5* (siRNA1 and siRNA2) and two negative controls (one targeting *GAPDH* and a

scrambled siRNA). The second and third columns show results from infection assay with PR8-GFP (H1N1, multiplicity of infection [MOI] of 0.4 to 10), with the latter column showing infectivity relative to the scrambled siRNA control. The *GAPDH* siRNA (but not the two siRNAs against *B3GALT5*) significantly reduced *GAPDH* expression relative to the scrambled siRNA (-90% reduction). *P*-values derived from a two-sided Wilcoxon Rank Sum Test and asterisks (*) mark those experiments with *P* < 0.05.

|---|---|

# Reporting Summary

## Statistics

For all statistical analyses, confirm that the following items are present in the figure legend, table legend, main text, or Methods section.

| n/a | Confirmed | |
|---|---|---|
| ☐ | ☒ | The exact sample size (*n*) for each experimental group/condition, given as a discrete number and unit of measurement |
| ☐ | ☒ | A statement on whether measurements were taken from distinct samples or whether the same sample was measured repeatedly |
| ☐ | ☒ | The statistical test(s) used AND whether they are one- or two-sided<br>*Only common tests should be described solely by name; describe more complex techniques in the Methods section.* |
| ☐ | ☒ | A description of all covariates tested |
| ☐ | ☒ | A description of any assumptions or corrections, such as tests of normality and adjustment for multiple comparisons |
| ☐ | ☒ | A full description of the statistical parameters including central tendency (e.g. means) or other basic estimates (e.g. regression coefficient) AND variation (e.g. standard deviation) or associated estimates of uncertainty (e.g. confidence intervals) |
| ☐ | ☒ | For null hypothesis testing, the test statistic (e.g. *F*, *t*, *r*) with confidence intervals, effect sizes, degrees of freedom and *P* value noted<br>*Give P values as exact values whenever suitable.* |
| ☒ | ☐ | For Bayesian analysis, information on the choice of priors and Markov chain Monte Carlo settings |
| ☒ | ☐ | For hierarchical and complex designs, identification of the appropriate level for tests and full reporting of outcomes |
| ☐ | ☒ | Estimates of effect sizes (e.g. Cohen's *d*, Pearson's *r*), indicating how they were calculated |

*Our web collection on statistics for biologists contains articles on many of the points above.*

## Software and code

Policy information about availability of computer code

| Data collection | Individual-level exome sequencing, genotype and phenotype data is available to approved researchers via UKB at: https://www.ukbiobank.ac.uk/enable-your-research. Influenza GWAS summary statistics from FinnGen Release 8 are available to approved individuals after accepting the terms and licenses of the data. Influenza A seropositivity from Scepanovic et al., were downloaded from the GWAS catalogue (accession #GCST006339).<br><br>Single-sample processing, all in DNAnexus<br>-Conversion of sequencing data in BCL format to FASTQ format and the assignments of paired-end sequence reads to samples based on 10-base barcodes; bcl2fastq v2.19.0 https://support.illumina.com/sequencing/sequencing_software/bcl2fastq-conversion-software.html<br>-Read alignment; bwa 0.7.17 http://bio-bwa.sourceforge.net<br>-Duplicate marking, stats gathering; picard v1.141 https://broadinstitute.github.io/picard/<br>-SAM/BAM/CRAM file generation and manipulation; samtools v1.7 http://www.htslib.org<br>-Variant calling; WeCall v1.1.2 https://github.com/Genomicsplc/wecall<br>-Sequence Quality Control; FastQC 0.11.8 http://www.bioinformatics/babraham.ac.uk/projects/fastqc/<br>-VCF file manipulation and index generation; bcftools v1.7 http://www.htslib.org, bgzip/tabix v1.7 http://www.htslib.org<br>-haplotyping (Ancestry.com); Eagle v2.4.1 https://github.com/poruloh/Eagle<br>-imputation (Ancestry.com): Minimac4 v1.01 https://github.com/statgen/Minimac4<br><br>Generation of "freeze" data<br>-Joint genotyping to generate project-level VCF (pVCF) files; GLnexus v1.4.5 https://github.com/dnanexus-rnd/GLnexus<br>-Generation of variant representations in PLINK format; PLINK v1.90b6.21 https://www.cog-genomics.org/plink2/ |
|---|---|

| Data analysis | -Ancestry predictions, IBD (Identity-by-descent) estimate, and pedigree reconstruction; PLINK v1.90b6.21 https://www.coggenomics.org/plink2/ |
|---|---|
| | - association testing: REGENIE v3.1.3 https://github.com/rgcgithub/regenie.<br>- meta-analysis: METAL (2020-05-05) https://github.com/statgen/METAL.<br>- various: python v3.8 https://www.python.org/downloads/; R v4.0.4 https://cran.r-project.org, R packages include ggplot2 (v3.4.2) and patchwork (v1.1.3). Python packages include pandas (v2.0.3) and numpy (v1.25.2).<br>- Plots for in vitro experiments: GraphPad Prism 9.3.0<br>- qPCR analysis: QuantStudio 6 (v2.6) |

For manuscripts utilizing custom algorithms or software that are central to the research but not yet described in published literature, software must be made available to editors and reviewers. We strongly encourage code deposition in a community repository (e.g. GitHub). See the Nature Portfolio guidelines for submitting code & software for further information.

# Data

Policy information about availability of data

All manuscripts must include a data availability statement. This statement should provide the following information, where applicable:
- Accession codes, unique identifiers, or web links for publicly available datasets
- A description of any restrictions on data availability
- For clinical datasets or third party data, please ensure that the statement adheres to our policy

Summary statistics from our GWAS will be made publicly available. Individual-level exome sequencing, genotype and phenotype data is available to approved researchers via UKB at: https://www.ukbiobank.ac.uk/enable-your-research. FinnGen Release 8 influenza GWAS summary statistics are available after accepting the terms and licenses. Influenza A seropositivity from Scepanovic et al., were downloaded from the GWAS catalogue (accession #GCST006339). Pre-calculated LD scores from the 1000 Genomes9 European reference population were obtained from https://data.broadinstitute.org/alkesgroup/LDSCORE/.

# Human research participants

Policy information about studies involving human research participants and Sex and Gender in Research.

| Reporting on sex and gender | Neither sex nor gender were considered in the study design. Analyses were not stratified by sex, although genetically determined sex was used as a covariate in the GWAS. |
|---|---|
| Population characteristics | Population characteristics (e.g., age, ancestry) can be found in Tables S1-2. |
| Recruitment | UK Biobank recruited approximately 500,000 individuals 40-69 years of age in 2006 to 2010 by mailers to people in the UK medical system (54.3% female). Informed consent was obtained for all participants. AncestryDNA customers over age 18, living in the United States, and who had consented to research, were invited to complete a survey assessing COVID-19 outcomes and other demographic information including SARS-CoV-2 swab and antibody test results, COVID-19 symptoms and severity, brief medical history, household and occupational exposure to SARS-CoV-2, and influenza infections (median age 57; 66.4% female). Geisinger Health System (GHS). The GHS MyCode Community Health Initiative is a health system-based cohort from central and eastern Pennsylvania (USA) with ongoing recruitment since 2006 (ages 19-94; 61.1% female). Penn Medicine BioBank (PMBB) study participants are recruited through the University of Pennsylvania Health System, which enrolls participants during hospital or clinic visits (ages 19-90; 50.7% female). Project Generation included 116,277 subjects from the Mayo Clinic Biobank (enrolled beginning in 2009) and 30 disease-specific registries (ages 19-98; 55.7% female). The ATLAS Precision Health Biobank at UCLA comprises ~32,000 individuals (ages 18-91; 55.7% female). De-identified phenotype data comprises all hospital visits beginning in 2013 and converted to ICD-10 codes. The CCPM biobank at the University of Colorado Anschutz Medical Campus in Aurora encompasses ~45,000 individuals (ages 30-92; 61.1% female). De-identified phenotype data was collected by Health Data Compass and comprises an individual's entire medical record from the University of Colorado's EHR. |
| Ethics oversight | Ethical approval for the UK Biobank was previously obtained from the North West Centre for Research Ethics Committee (11/NW/0382). The work described herein was approved by UK Biobank under application number 26041. GHS study: approval for DiscovEHR analyses was provided by the Geisinger Health System Institutional Review Board (#2006-0258). AncestryDNA study: all data for this research project was from subjects who provided prior informed consent to participate in AncestryDNA's Human Diversity Project, as reviewed and approved by our external IRB (Pro00034516), Advarra. All data was de-identified prior to use. PMBB study: appropriate consent was obtained from each participant regarding storage of biological specimens, genetic sequencing and genotyping, and access to all available EHR data. This study was approved by the Institutional Review Board of the University of Pennsylvania and complied with the principles set out in the Declaration of Helsinki. Mayo-RGC Project Generation: all subjects provided informed consent for use of specimens and data in genetic and health research and ethical approval for Project Generation was provided by the Mayo-Clinic IRB (#09-007763). CCPM Biobank: ethical approval and consent was reviewed and approved by the Colorado Multiple Institutional Review Board (#15-0461). UCLA: patient recruitment and sample collection for Precision Health Activities at UCLA is an approved study by the UCLA IRB (#17-001013). Informed consent was obtained for all study participants. |

Note that full information on the approval of the study protocol must also be provided in the manuscript.

# Field-specific reporting

Please select the one below that is the best fit for your research. If you are not sure, read the appropriate sections before making your selection.

☒ Life sciences ☐ Behavioural & social sciences ☐ Ecological, evolutionary & environmental sciences

For a reference copy of the document with all sections, see nature.com/documents/nr-reporting-summary-flat.pdf

# Life sciences study design

All studies must disclose on these points even when the disclosure is negative.

| | |
|---|---|
| Sample size | Sample sizes were all those available in the individual cohorts as described in the text. No power calculations were performed or required in advance. |
| Data exclusions | Prior to any analysis, we established the following data exclusions: We excluded individuals that were not predicted to belong to 5 continental ancestry groups (AFR, EAS, EUR, HLA, SAS) and furthermore did not analyze sets of individuals with fewer than 100 cases and 100 controls. |
| Replication | Of the 2 GWAS loci discovered in AncestryDNA, we successfully replicated both in a separate meta-analysis including FinnGen, UKB, UPENN-PMBB, GHS, Mayo Clinic, UCLA and Colorado. In vitro infection assays were successfully repeated at least twice. |
| Randomization | We performed a GWAS, which was an observational study, and as such no process of randomization was performed or applicable here because there was no allocation of samples into experimental groups. |
| Blinding | We performed a GWAS, which was an observational study, using coded de-identified data. As such, no process of blinding to group allocation was performed or applicable here. |

# Reporting for specific materials, systems and methods

We require information from authors about some types of materials, experimental systems and methods used in many studies. Here, indicate whether each material, system or method listed is relevant to your study. If you are not sure if a list item applies to your research, read the appropriate section before selecting a response.

## Materials & experimental systems

| n/a | Involved in the study |
|---|---|
| ☐ | ☒ Antibodies |
| ☐ | ☒ Eukaryotic cell lines |
| ☒ | ☐ Palaeontology and archaeology |
| ☒ | ☐ Animals and other organisms |
| ☒ | ☐ Clinical data |
| ☒ | ☐ Dual use research of concern |

## Methods

| n/a | Involved in the study |
|---|---|
| ☒ | ☐ ChIP-seq |
| ☒ | ☐ Flow cytometry |
| ☒ | ☐ MRI-based neuroimaging |

## Antibodies

| | |
|---|---|
| Antibodies used | anti-ST6GAL1 antibody (goat, #AF5924, R&D system)<br>Chicken anti-goat HRP-conjugated antibody (#HAF019, R&D system)<br>Beta-actin HRP (#5123, Cell Signaling)<br>GAPDH HRP (#HRP-60004, Proteintech) |
| Validation | anti-ST6GAL1 antibody detects human ST6 Gal Sialyltransferase 1/ST6GAL1 in direct ELISAs and Western blots and has been cited in 23 publications. Both we and the vendor report reduction of the signal following siRNA-mediated knockdown. Chicken anti-goat HRP-conjugated antibody Detects goat IgG heavy and light chains in direct ELISAs and Western blots. In Western blots, less than 5% cross-reactivity with mouse IgG, rabbit IgG and human IgG is observed. It has been used in 23 publications. Both Beta-actin HRP and GAPDH HRP are common loading controls antibodies and they have been used in hundreds of publications. |

## Eukaryotic cell lines

Policy information about cell lines and Sex and Gender in Research

| | |
|---|---|
| Cell line source(s) | A549 (CCL-185) and Calu-3 (HTB-55) cells were purchased from ATCC. |
| Authentication | Authentication via STR analysis was provided by the vendor (ATCC). |

Mycoplasma contamination | Cultured cells were tested monthly for mycoplasma contamination using the Lonza MycoAlert Kit and tested negative.

Commonly misidentified lines
(See ICLAC register) | No commonly misidentified cells were used in this study.

