## [Peer Review File · Nature Genetics]

Peer Review Information

Manuscript Title: Genetic risk factors for COVID-19 and influenza are largely distinct

Corresponding author name(s): Dr Manuel (AR) Ferreira, Dr Gonçalo (R) Abecasis

Reviewer Comments & Decisions:

Decision Letter, initial version:
--

11th October 2022

Dear Manuel,

Your Article "Analysis of common variants shows that risk factors for COVID-19 are largely distinct from flu and identify first confirmed associations with risk of influenza infection" has been seen by three referees. You will see from their comments below that, while they find your work of potential interest, they have raised substantial overlapping concerns that must be addressed. In light of these comments, we cannot accept the manuscript for publication at this time, but we would be interested in considering a suitably revised version that addresses the key concerns.

We hope you will find the referees' comments useful as you decide how to proceed. If you wish to submit a substantially revised manuscript, please bear in mind that we will be reluctant to approach the referees again in the absence of major revisions.

To guide the scope of the revisions, the editors discuss the referee reports in detail within the team, including with the chief editor, with a view to identifying key priorities that should be addressed in revision, and sometimes overruling referee requests that are deemed beyond the scope of the current study. In this case, we agree with the referees that it would be advisable to restructure the paper to focus primarily on the genome-wide association analyses of influenza infection, rather than comparative analyses of influenza and COVID-19 host susceptibility factors. In addition to this major restructuring, we ask that you provide more detailed discussion of how case and control definitions and ascertainment criteria might influence the association findings and their interpretation and extend the functional studies of ST6GAL1 as recommended by Reviewer #3. We hope you will find this prioritized set of referee points to be useful when revising your study. Please do not hesitate to get in touch if you would like to discuss these issues further.

If you choose to revise your manuscript taking into account all reviewer and editor comments, please highlight all changes in the manuscript text file. At this stage we will need you to upload a copy of the manuscript in MS Word .docx or similar editable format.

*2) If you have not done so already please begin to revise your manuscript so that it conforms to our Article format instructions, available here. Refer also to any guidelines provided in this letter.

Please be aware of our guidelines on digital image standards.

[redacted]

If you wish to submit a suitably revised manuscript we would hope to receive it within 3-6 months. If you cannot send it within this time, please let us know. We will be happy to consider your revision so long as nothing similar has been accepted for publication at Nature Genetics or published elsewhere. Should your manuscript be substantially delayed without notifying us in advance and your article is eventually published, the received date would be that of the revised, not the original, version.

Thank you for the opportunity to review your work.

Sincerely,
Kyle

Kyle Vogan, PhD
Senior Editor
Nature Genetics
<https://orcid.org/0000-0001-9565-9665>

Referee expertise:

Referee #1: Genome-wide association, infectious diseases, population genetics

Referee #2: Genome-wide association, infectious diseases, population genetics

Referee #3: Genome-wide association, infectious diseases, virology

Reviewers' Comments:

Reviewer #1:
Remarks to the Author:

The authors compare the genetic architecture of influenza and COVID-19 infection through a GWAS comparing 18,334 cases and ~276k controls for influenza.

It is unclear why the authors would expect the two infections to have shared genetic architecture given that the introduction states they have distinct mechanisms. It is not novel that different pathogens have different mechanisms and therefore different genetics underlying their pathogenesis.

My primary concern is phenotypic definition, including considering these cases to be representative of all infection and controls as true controls. In order to make any conclusions about susceptibility to infection, both cases and controls would need to have the chance to develop the outcome. With COVID-19, this was already tenuous. With influenza, it is even more unrealistic that every control was exposed to the virus, especially given influenza's relatively low rates in the past couple of years due to COVID-19 mitigation strategies. Without precision in the phenotype definitions, the analysis can be subject to confounding and the conclusions called into question. There is also notable selection bias in who gets tested for influenza which may influence results. At the very least, a more extensive discussion of these concerns is warranted in the limitations for this study.

Overall, the strength of this manuscript comes from focusing specifically on influenza, not necessarily infection but symptomatic illness, likely severe if testing was warranted. The knockdown experiments to verify this relationship also strengthen the manuscript. It is unclear why the authors sought to bring COVID-19 into this at all. It doesn't belong and if anything raises questions due to the phenotypic relationship between the two outcomes in this sample. I would advise the authors to drop these

components and focus on the strongest aspects of this work, which is centering influenza.

Some specific comments are below:

1. The authors note an inverse relationship between self-reported influenza and COVID-19 infection. They make conclusions that this indicates that the two diseases show few common genetic variants, but also acknowledge that this may be due to the sampling through the survey. A more in-depth exploration of why this would exist is warranted instead of inferring biological conclusions from potential flaws in study design.
2. The authors conclude that a modest genetic correlation between COVID-19 hospitalization and self-reported influenza indicates some sharing but substantial divergence of genetic etiology between the two traits. Given the relationship between the phenotypes, it is difficult to see how they could make such conclusions. It could be because of the underlying genetics. It could also be because those who were hospitalized may have already been in poor health for a different reason than COVID-19 which would also lead for them to have been tested for influenza in the first place.
3. Again, the authors state that "Given multiple axes of similarity, the observation that COVID-19 risk variants identified to date did not have a significant association with susceptibility to influenza – individually or in aggregate – was unexpected." This should not be unexpected. Many many many pathogens with similar presentation from respiratory infections to parasitic infections to enteric infections have shown distinct genetic profiles without sharing genetic variants. The authors would benefit from reviewing infectious disease literature.

Reviewer #2:

Remarks to the Author:

I read Kosmicki J. et al. with great interest, and congratulate the authors on the strong results identifying a key influenza susceptibility gene. While the authors report several interesting findings, they also cover a lot of territory, switching gears multiple times and failing to address some of the shortcomings of the data. Starting with a comparative genomic analysis between influenza and COVID-19, they validate a previously reported protective loci for influenza, evaluate the role of rare variants in influenza risk, conduct in vitro experiments to reduce expression of ST6GAL1, and show how interference with siRNA of this gene confers reduced influenza infectivity. Genetic and phenotypic data were obtained from AncestryDNA's COVID study, Geisinger health system DiscovEHR study, and UK Biobank for analysis. GWAS were generated for COVID-19 and influenza infection. Leveraging results published from the COVID HGI, genetic correlation was evaluated between SARS-CoV-2 infection and hospitalization using LD score regression.

Regarding the comparative analysis between influenza infection and COVID-19 infection, the most relevant locus appears to be ABO, which is dismissed due to results from secondary analysis of COVID-19 contaminated controls. However, this same pattern was described previously using influenza data from 2017 and 2018 (pre-COVID-19 pandemic) (<https://www.nature.com/articles/s41588-021-00854-7>), with opposite directionality at the ABO locus between COVID-19 and influenza.

It is unfortunate that self-reported infection is the only variable showing significant results, but due to the lack of testing for influenza, it begs the question of who gets tested for influenza and why. How is this related to severity? Are you examining a more severe population with your test positive group compared to self-report? Finally, examining the shared genetic architecture during the same time period as the pandemic is potentially unnecessarily complicating things due to inadequate testing for both conditions. Please review the above reference again, and appropriately contextualize the ABO locus in the comparative analysis between COVID-19 and influenza given prior findings. Also, please describe precisely how influenza status was obtained in the AncestryDNA questionnaire.

Given that your interest is in illuminating the shared architecture of these traits, it seems an MTAG analysis (<https://pubmed.ncbi.nlm.nih.gov/29292387/>) would be ideal, however that approach was not chosen. Please describe in one sentence the rationale for your approach, given the MTAG may help identify some of those sub-threshold associations that were not significant in the original results.

To further probe the connection between influenza and COVID-19, a genetic risk score was generated based on the significant associations, which did not discriminate between influenza cases and controls. This analysis does not contribute to the paper in any way, nor would it be expected to, and fails to conclusively provide information one way or the other. I recommend striking this element of the paper completely.

The correlation between infection with influenza and SARS-CoV-2 is underwhelming, but with SARS-CoV-2 hospitalization is higher. Is there any way for you to correlate severity of infection across both infections? In large part, we are not collectively trying to treat infection per se, but are trying to prevent severe disease which is where the therapeutic potential is. If influenza severity is unavailable, please describe that as a shortcoming to this paper's ability to fully examine the question of shared genetic architecture between influenza and COVID-19.

I will stop there as my expertise is not in the functional characterization of genes, and let others weigh in. I look forward to seeing this paper in a final state, and again, congratulate the authors on a job well done.

Reviewer #3:
Remarks to the Author:

Considering that respiratory single-stranded RNA viruses cause both SARS-CoV-2 infection and flu, the authors wondered if these two infections share genetic susceptibility. The group is very strong in genetic analyses, which are proper in general. However, this approach does not seem logical for an extremely common, mostly mild and recurrent infection ascertained during a major pandemic with extensive masking decreasing the transmission of any respiratory infection.

There is more precise ascertainment for SARS-CoV-2 infection, a novel and still not very common infection in which infection status is reasonably clear (yes vs. no/not yet). I am struggling to apply this logic to flu. The cases were defined as those with positive self-reported flu test (during April 2020 – February 2021) vs. those with negative test or no test. I personally and basically anyone I can think of had a respiratory infection (flu?) multiple times in my life but never had a formal test – would I be considered in the control group? Asking for controls who never had a flu infection (not just a positive

flu test) would be a more appropriate control group, but I doubt any controls would be found. Unfortunately, instead of genetic susceptibility to flu, the study attempts to identify genetic factors for having a positive test during April 2020 – February 2021 vs. a very broad and poorly defined group of controls, which is quite meaningless for identifying biological mechanisms.

It's no wonder there are no GWAS regions reported for flu so far and an analysis of 10 million common variants in 18,334 cases with a positive test for influenza and 276,295 controls is unlikely to help. The authors conclude that "the genetic architectures of COVID-19 and influenza are largely distinct," which is greatly influenced by the selection of controls.

The analyses of rare variants were affected by the same ascertainment issues. This issue is not even brought up in the discussion. The authors picked a very difficult/unreliable phenotype for their analysis and the negative results are what could be expected.

The associations between flu and common variants in ST6GAL1 and B3GALT5 are the only new, very limited and still preliminary data. Only transient siRNA-KD for ST6GAL1 resulted in some moderate effect on flu infectivity in vitro. Figure 2 shows a noticeable effect of GAPDH siRNA on ST6GAL1 expression. Similarly, Fig S13 shows an effect of GB3GALT siRNA on GAPDH expression. All significant p-values should be on the plots. Perhaps more optimization of these experiments is required and the effects of these siRNAs on protein expression of the targets need to be demonstrated by Western blots.

The effects of putatively associated genetic variants on corresponding proteins and on flu infection should be demonstrated by multiple additional methods. For example, estimating in-vitro infectivity in primary epithelial cells from individuals with different genotypes or in primary cells CRISPR-edited for these genotypes.

Figure 1. rs numbers would be very informative on the plot.

Author Rebuttal to Initial comments

Reviewer #1

1. It is unclear why the authors would expect the two infections to have shared genetic architecture given that the introduction states they have distinct mechanisms. It is not novel that different pathogens have different mechanisms and therefore different genetics underlying their pathogenesis.

Response: We do agree that there are unique mechanisms underlying infection with SARS-CoV-2 and influenza, for example the cell surface molecules used as attachment factors to infect human cells (ACE2 for SARS-CoV-2 and sialic acid residues for influenza). However, we postulated that COVID-19 and influenza could also have shared genetic risk factors because both diseases partially share clinical risk factors, some underlying molecular pathways (e.g., interferon response) and symptoms. We believe our results – specifically (i) the lack of association between top loci for COVID-19 and influenza, and vice-versa; and (ii) the modest but statistically significant

genetic correlation estimated between the two diseases ($r_g \sim 0.3$) – support the notion that most but not all genetic risk factors are unique to each disease.

2. *My primary concern is phenotypic definition, including considering these cases to be representative of all infection and controls as true controls. In order to make any conclusions about susceptibility to infection, both cases and controls would need to have the chance to develop the outcome. With COVID-19, this was already tenuous. With influenza, it is even more unrealistic that every control was exposed to the virus, especially given influenza's relatively low rates in the past couple of years due to COVID-19 mitigation strategies. Without precision in the phenotype definitions, the analysis can be subject to confounding and the conclusions called into question. There is also notable selection bias in who gets tested for influenza which may influence results. At the very least, a more extensive discussion of these concerns is warranted in the limitations for this study.*

Response: The reviewer highlights important limitations of the influenza phenotype analyzed in the AncestryDNA cohort, namely that (i) cases are not representative of all infection; and (ii) controls include individuals who were not exposed to influenza (and so did not have a chance to develop flu) or were exposed but not tested. We now explicitly state these limitations and performed additional analyses to understand the extent to which the two associations with flu are robust to phenotype definition. Specifically:

- 1) On page 4, we highlight the limitations of the case-control definition used in the AncestryDNA discovery analysis. In this analysis, we compared individuals with (cases) against those without (controls) a positive test for influenza across the 2019-20 or 2020-21 flu seasons. As the reviewer points out, we do not know what proportion of controls were exposed to the virus in either season and had the chance of developing flu. This would be very difficult (if not impossible) to establish in a large cohort. Such misclassification will reduce power to discover genetic risk factors for susceptibility to infection, which is partially offset by the large sample size. We state:

“We refer to this phenotype as “reported influenza infection”, but recognize that it does not represent true susceptibility to infection, because the control group includes an undetermined number of individuals who were not exposed to influenza in either season (and so did not have a chance to develop flu) or who were exposed and infected but not tested (e.g., asymptomatic). As such, this phenotype may represent risk of having a symptomatic influenza infection that required seeking (or being prescribed) a viral test.”

- 2) We present association results from 8 additional influenza phenotypes (from AncestryDNA, EHR cohorts and published literature) representing both looser and stricter definitions, to assess the extent to which the two flu associations are robust to phenotype

definition. Results from these analyses are summarized in the new Figure 2 and the full details can be found in Supplementary Table 2. Briefly, we found that:

- a. In the AncestryDNA cohort (discovery), the effect size for both loci remained the same when we restricted the control group to 23,985 individuals (down from 276,295) who reported being tested for influenza and having a negative result: OR=0.86 (vs. 0.86) and $P=5.2 \times 10^{-6}$ for *ST6GALI*, and OR=0.89 (vs. 0.90) and $P=4.9 \times 10^{-12}$ for *B3GALT5*. That is, analyzing a stricter phenotype that considers only individuals who were tested for influenza (minimizing the potential selection bias highlighted by the reviewer) did not impact the results.
- b. In the AncestryDNA cohort, the effect size for both loci was attenuated when analyzing a looser influenza phenotype that ignores information from influenza test results and instead defines cases and controls based solely on the presence of flu-like symptoms (44K cases vs. 250K controls): OR=0.93 (vs. 0.86) and $P=1.7 \times 10^{-7}$ for *ST6GALI*, and OR=0.95 (vs. 0.90) and $P=4.0 \times 10^{-11}$ for *B3GALT5*. However, both loci remained highly associated given the large case sample size.
- c. In the GHS cohort, which is EHR based, we were able to extract results from cell-based viral (influenza A and B) culture assays for 82K individuals. Typically, these screening assays are requested when a patient presents with symptoms consistent with upper respiratory infections. A positive test indicates a current infection. We found that both variants were significantly associated with lower risk of having a positive test for influenza A (1,694 cases vs. 80,434 controls): OR=0.86 and $P=3 \times 10^{-5}$ for *B3GALT5*; OR=0.82 and $P=0.005$ for *ST6GALI*. Consistent results were observed for influenza B, although there were only 235 cases: OR=0.75 and $P=0.002$ for *B3GALT5*; OR=0.89 and $P=0.51$ for *ST6GALI*. Therefore, among individuals presenting with symptoms leading to a viral test in a medical setting, both variants show a consistent association with protection from influenza infection.
- d. In the GHS and UKB cohorts, we were able to separate influenza cases into two groups: with (N=2,951) vs. without (N=6,984) a hospital admission for flu in their available medical records. We only considered records with influenza listed as a primary cause of hospitalization. Based on these data, we found that the *B3GALT5* variant was associated with 10% lower risk of hospitalization (OR=0.898 and $P=0.004$). There was no association with the *ST6GAL1* variant (OR=0.955 and $P=0.52$).
- e. Lastly, we found a small population-based study (PMID 30053915, 25562703) that measured IgG antibodies against influenza A in 1,000 healthy individuals from

France, in 2012-13 (50% females, uniformly distributed between ages of 20 and 70, no evidence of severe/chronic/recurrent medical conditions). The authors then performed a GWAS comparing 777 seropositive cases against 223 seronegative controls, with genome-wide summary statistics deposited in the GWAS catalog. In this study, both variants were associated with lower risk of having a positive serology test for influenza, significantly so for B3GALT5: OR=0.696 and P=0.001.

These findings demonstrate that both associations are relatively robust to phenotype definition, with a trend for effect sizes to be stronger with stricter (more specific) definitions of influenza infection.

3. Overall, the strength of this manuscript comes from focusing specifically on influenza, not necessarily infection but symptomatic illness, likely severe if testing was warranted. The knockdown experiments to verify this relationship also strengthen the manuscript. It is unclear why the authors sought to bring COVID-19 into this at all. It doesn't belong and if anything raises questions due to the phenotypic relationship between the two outcomes in this sample. I would advise the authors to drop these components and focus on the strongest aspects of this work, which is centering influenza.

Response: We appreciate and to some extent share the reviewer's reservations on including COVID-19 data in this manuscript. However, the Regeneron-AncestryDNA collaboration was specifically designed to identify risk factors for COVID-19. A key question both teams wanted to address was whether some genetic risk factors for COVID-19 were shared with influenza, and so we included flu-related questions in the COVID-19 survey sent out to AncestryDNA participants who consented to research. As we briefly mention at the beginning of the manuscript, the reason why this question was important to us is because genetic risk factors shared between COVID-19 and flu (if any) could potentially point to targets that we would prioritize for therapeutic development over targets that are unique to each disease (assuming comparable tractability).

Arguments can be made either way over the validity of our hypothesis that some genetic risk factors are shared between these two conditions (yes, many mechanisms are distinct, but some are shared). Having influenza data in hand for such a large sample size, especially with replicable genome-wide associations, we were uniquely positioned to objectively test this hypothesis.

We hope this explains in more detail why we included COVID-19 data in this manuscript. We revised the following paragraph to briefly include this information:

"To understand the extent to which the same host genetic factors influence risk of COVID-19 and influenza, we first performed a GWAS of influenza infection based on survey data provided by 296,313 participants of the AncestryDNA COVID-19 study who consented to research [7].

Although the focus of that study was on risk factors for COVID-19, participants also indicated if they were tested for influenza in either the 2019-20 or 2020-21 flu seasons (Methods).”

Furthermore, we agree that the inclusion of the COVID-19 data in the manuscript made it both harder to read and highlight the key novel findings. For this reason, COVID-19-related sections were either removed (e.g., merged COVID-19 & influenza phenotype), considerably shortened or moved to the Supplement. We also changed the format to a Brief Communication, which we felt was a better fit to describe our findings concisely.

We hope these changes improve the readability of our manuscript.

4. The authors note an inverse relationship between self-reported influenza and COVID-19 infection. They make conclusions that this indicates that the two diseases show few common genetic variants, but also acknowledge that this may be due to the sampling through the survey. A more in-depth exploration of why this would exist is warranted instead of inferring biological conclusions from potential flaws in study design.

Response: We noted an inverse relationship between influenza and COVID-19 in the AncestryDNA study, but we did not state that this indicates that both diseases share few genetic risk factors. Instead, we postulated that this inverse relationship may contribute to the observation that the *ABO* variant that protects against COVID-19 was associated with higher risk of influenza.

5. The authors conclude that a modest genetic correlation between COVID-19 hospitalization and self-reported influenza indicates some sharing but substantial divergence of genetic etiology between the two traits. Given the relationship between the phenotypes, it is difficult to see how they could make such conclusions. It could be because of the underlying genetics. It could also be because those who were hospitalized may have already been in poor health for a different reason than COVID-19 which would also lead for them to have been tested for influenza in the first place.

Response: The reviewer raises the possibility that a modest positive genetic correlation between COVID-19 hospitalization and influenza may arise because individuals with poor health have a higher risk of both (i) being hospitalized for COVID-19 and (ii) being tested for influenza (and having a positive test), relative to individuals in good health. We find this unlikely, because risk variants for COVID-19 are typically not associated with clinical risk factors for COVID-19, such as type-2 diabetes, as we have shown previously (see Supp Table 8 in Horowitz et al. Nat Genet 2021, PMID 35241825).

Nonetheless, to objectively address this possibility, we estimated the genetic correlation between COVID-19 and influenza using results from our analysis of medical record influenza in the EHR cohorts. In these cohorts, there is minimal (or no) relationship between the two phenotypes, because most diagnoses of influenza were given before the COVID-19 pandemic (92% of

influenza cases had a record of influenza infection prior to 2020). In this analysis, the genetic correlation between COVID-19 phenotypes and medical record influenza was also modest (see Table below). To improve precision of the genetic correlation estimates, we meta-analyzed the discovery (reported positive test) and replication (medical record) influenza GWAS and then re-estimated the genetic correlation with COVID-19 phenotypes. In this analysis, we again observed modest genetic correlations between the two diseases ($r_g \sim 0.3$). We have updated Supp Table 5 to include these results.

Influenza trait	COVID-19 trait (HGI)	r_g (SE)	P-value
Reported positive influenza test (AncestryDNA)	SARS-CoV-2 infection	0.211 (0.138)	0.1270
	COVID-19 hospitalization	0.391 (0.137)	0.0044
Medical record influenza (EHR cohorts)	SARS-CoV-2 infection	0.411 (0.190)	0.0304
	COVID-19 hospitalization	0.134 (0.167)	0.4222
Meta-analysis of AncestryDNA and EHR cohorts	SARS-CoV-2 infection	0.343 (0.126)	0.0065
	COVID-19 hospitalization	0.304 (0.117)	0.0091

6. Again, the authors state that "Given multiple axes of similarity, the observation that COVID-19 risk variants identified to date did not have a significant association with susceptibility to influenza – individually or in aggregate – was unexpected." This should not be unexpected. Many many pathogens with similar presentation from respiratory infections to parasitic infections to enteric infections have shown distinct genetic profiles without sharing genetic variants. The authors would benefit from reviewing infectious disease literature.

Response: We appreciate the reviewer's point, but as we mentioned above, we felt there were sufficient known risk factors, physiological mechanisms and symptoms shared between the two diseases to justify raising (and designing a study to test) the hypothesis that some genetic risk factors would be shared. The other point to make is that genetic analyses of many infectious diseases have not been performed in large enough studies to have adequate power to identify genetic risk factors, aside from those with very large effect sizes (e.g., HLA). This, of course, then limits the ability to determine the extent to which different infectious diseases share genetic risk factors.

We did take the reviewer's advice and looked both in the literature and in our own data for evidence of shared genetic risk factors between different infectious diseases. We found a handful of non-HLA examples, as listed below:

- Variants in *FUT2* are associated with different infectious diseases, for example norovirus (PMID 16306606), mumps virus (PMID 28928442), ear infections (PMID 28928442), tonsillectomy (PMID 28928442), COVID-19 (Ganna et al., *medRxiv* 2022) and human polyomavirus (our unpublished analysis of serology data available in UKB [N~10K]). For

some of these diseases, the genetic signal is exactly the same. For example, the lead variant that increases risk of mumps (rs516316:C) is also a lead variant that increases antibody levels against human polyomavirus ($P=4.5 \times 10^{-20}$) and herpes simplex ($P=10^{-5}$) in the UKB.

- In our analysis of serology data in the UKB (N~10K), we further found that:
 - A variant located in a cluster of genes that encode the variable region of the heavy chain of immunoglobulins (rs2337939:G), is associated with higher antibody levels against human papillomavirus (rs2337939, $P=8 \times 10^{-9}$), T-lymphotropic virus 1 ($P=10^{-6}$), hepatitis C virus ($P=3 \times 10^{-5}$) and HIV ($P=0.001$).
 - A variant located near PTPRG (rs148128158:C) was associated with higher antibody levels against hepatitis B virus ($P=9 \times 10^{-9}$), HIV ($P=5 \times 10^{-5}$) and human papillomavirus ($P=6 \times 10^{-5}$).

These findings support the notion that some genetic risk factors may be shared between infectious diseases, although we agree that many are likely to be unique.

Reviewer #2:

1. Regarding the comparative analysis between influenza infection and COVID-19 infection, the most relevant locus appears to be ABO, which is dismissed due to results from secondary analysis of COVID-19 contaminated controls. However, this same pattern was described previously using influenza data from 2017 and 2018 (pre-COVID-19 pandemic) (<https://www.nature.com/articles/s41588-021-00854-7>), with opposite directionality at the ABO locus between COVID-19 and influenza. It is unfortunate that self-reported infection is the only variable showing significant results, but due to the lack of testing for influenza, it begs the question of who gets tested for influenza and why. How is this related to severity? Are you examining a more severe population with your test positive group compared to self-report? Finally, examining the shared genetic architecture during the same time period as the pandemic is potentially unnecessarily complicating things due to inadequate testing for both conditions. Please review the above reference again, and appropriately contextualize the ABO locus in the comparative analysis between COVID-19 and influenza given prior findings.

Response: Thank you for highlighting this paper (Shelton et al. 2021, PMID 33888907) and their original observation that the ABO locus has opposing effects on risk of COVID-19 and influenza. Our initial submission did briefly note this observation (and cited this paper), but we did not dissect it further. Reading this paper in more detail, we note the following:

- First, the lead variant for COVID-19 at the ABO locus in that paper is rs9411378 ($P=5 \times 10^{-20}$, OR = 0.86 for the C allele). The authors note that this variant is in LD ($r^2=0.57$) with a frameshift variant in ABO (rs8176719) that determines the O blood group when present in the homozygous form. We were skeptical of this connection to O blood group type given the modest LD, but have confirmed that the ABO variant reported in the larger HGI study (and that we tested for association with influenza; rs505922) is indeed in high LD with the ABO frameshift variant ($r^2=0.89$).
- The authors then go on to show that individuals with O blood group have lower risk of COVID-19 (OR~0.8) but higher risk of flu in 2017 and 2018 (OR~1.05). This is consistent with our results. The reviewer makes the excellent point that the association with influenza in this study cannot be explained by a contamination of COVID-19 cases among the influenza controls, because there was no COVID-19 in 2017-18.

So, the question remains as to why the ABO locus is associated with both influenza and COVID-19 in both the Shelton et al. and our analyses. We see a consistent association between the COVID-19 variant rs505922 and influenza in both our discovery analysis ($P=2.18 \times 10^{-4}$; OR=1.046) and ICD10 replication ($P=0.0084$; OR=1.027), with an overall meta-analysis OR=1.04, $P=1.1 \times 10^{-5}$.

However, rs505922 is not the lead influenza-associated variant at this locus, instead being rs2519093 (OR=1.05; $P=1.4 \times 10^{-6}$). There is only modest LD between rs2519093 and rs505922 ($r^2=0.48$), indicating that the top influenza signal in the *ABO* locus does not co-localize with the COVID-19 GWAS signal.

Furthermore, we note that many diseases and quantitative traits are associated with variants in the *ABO* locus, of which we highlight the following infectious diseases:

- 1) Childhood ear infections [PMID 28928442; rs8176643, OR=1.06, $P=3.7 \times 10^{-11}$],
- 2) Malaria [PMID 31844061; rs8176751, OR=1.20, $P=6.4 \times 10^{-10}$]
- 3) Tonsillectomy [PMID 28928442; rs635634, OR=1.06, $P=8.47 \times 10^{-9}$].

The relationship between these associations and the lead influenza variant is shown below:

These results suggest that the influenza signal at the *ABO* locus – if it is a true positive finding – co-localizes with the ear infection and tonsillectomy associations, but not COVID-19 or malaria. We now discuss this locus in greater detail in the Supplementary Information.

2. Also, please describe precisely how influenza status was obtained in the AncestryDNA questionnaire.

Response: Apologies if this was not clear in the original submission. We had included this information in Supplementary Table 13 (now Supplementary Table 10), but now also describe how influenza status was defined in the Methods section.

3. Given that your interest is in illuminating the shared architecture of these traits, it seems an MTAG analysis (<https://pubmed.ncbi.nlm.nih.gov/29292387/>) would be ideal, however that

approach was not chosen. Please describe in one sentence the rationale for your approach, given the MTAG may help identify some of those sub-threshold associations that were not significant in the original results.

Response: Following advice from the reviewers and editors, we minimized the number of COVID-19-related analyses included in the revision, including only the absolutely essential components. This section has now been removed.

4. To further probe the connection between influenza and COVID-19, a genetic risk score was generated based on the significant associations, which did not discriminate between influenza cases and controls. This analysis does not contribute to the paper in any way, nor would it be expected to, and fails to conclusively provide information one way or the other. I recommend striking this element of the paper completely.

Response: We appreciate the reviewer's suggestion and removed this analysis from the manuscript.

5. The correlation between infection with influenza and SARS-CoV-2 is underwhelming, but with SARS-CoV-2 hospitalization is higher. Is there any way for you to correlate severity of infection across both infections? In large part, we are not collectively trying to treat infection per se, but are trying to prevent severe disease which is where the therapeutic potential is. If influenza severity is unavailable, please describe that as a shortcoming to this paper's ability to fully examine the question of shared genetic architecture between influenza and COVID-19.

Response: We fully agree with the reviewer. As part of this revision, we worked extensively with UKB and GHS data to extract influenza information from in-patient hospital records. Specifically, we identified individuals in UKB and GHS with influenza listed as the primary cause for hospitalization. Based on this information, we defined the following severity phenotypes:

- Hospitalized influenza vs. population controls (1,696 cases vs. 593,185 controls)
- Not hospitalized influenza vs. population controls (6,984 cases vs. 593,185 controls)
- Hospitalized vs. not hospitalized influenza (1,696 cases vs. 6,984 controls)

The association between the two influenza loci (*ST6GAL1* and *B3GALT5*) and these phenotypes is now included in Supplementary Table 2 and a new Figure 2, which is shown below. Briefly, the *B3GALT5* variant was significantly associated with lower risk of hospitalization among cases (OR=0.88, P=0.005), with a comparable albeit non-significant association with the less common *ST6GAL1* variant (OR=0.89, P=0.17).

Reviewer #3

1. Considering that respiratory single-stranded RNA viruses cause both SARS-CoV-2 infection and flu, the authors wondered if these two infections share genetic susceptibility. The group is very strong in genetic analyses, which are proper in general. However, this approach does not seem logical for an extremely common, mostly mild and recurrent infection ascertained during a major pandemic with extensive masking decreasing the transmission of any respiratory infection. There is more precise ascertainment for SARS-CoV-2 infection, a novel and still not very common infection in which infection status is reasonably clear (yes vs. no/not yet). I am struggling to apply this logic to flu. The cases were defined as those with positive self-reported flu test (during April 2020 – February 2021) vs. those with negative test or no test. I personally and basically anyone I can think of had a respiratory infection (flu?) multiple times in my life but never had a formal test – would I be considered in the control group? Asking for controls who never had a flu infection (not just a positive flu test) would be a more appropriate control group, but I doubt any controls would be found. Unfortunately, instead of genetic susceptibility to flu, the study attempts to identify genetic factors for having a positive test during April 2020 – February 2021 vs. a very broad and poorly defined group of controls, which is quite meaningless for identifying biological mechanisms.

Response: We thank the reviewer for their comments, which we completely understand and considered often throughout this project. We fully agree that the primary influenza phenotype analyzed in the AncestryDNA cohort (reported positive test for influenza vs. negative or no test for influenza) is noisy, difficult to interpret and includes misclassified controls and a heterogeneous set of cases. The question for us was whether this phenotype was useful for genetic studies that seek to identify host factors that influence risk of having an overt influenza infection, that is, an infection that required testing or medical follow-up.

We believe this is the case because we were able to show that the two loci discovered in AncestryDNA had consistent and significant associations with more objective measures of influenza infection, such as:

- Lifetime medical record of influenza infection (ie. was a person diagnosed with flu in a medical setting at any point in their life). In this revision, we expanded our replication analysis of EHR cohorts to include three additional studies (UCLA, Colorado, Mayo-Clinic). Both loci are now genome-wide significant in these replication cohorts alone;
- Phenotypes that capture recent influenza infection, for example a positive cell culture for influenza A (a new phenotype we obtained from the GHS cohort) or positive serology test for influenza A (based on a published GWAS).

The new Figure 2 (included below) summarizes the associations between all influenza-related phenotypes tested and both loci discovered in the AncestryDNA cohort.

Regarding the reviewer's question, yes, anyone who had flu throughout their life but never got tested or sought medical treatment would be considered a control in our AncestryDNA or EHR cohorts. Presumably such individuals either had asymptomatic influenza or symptoms that were not severe enough to require medical follow-up. Our interpretation is that the phenotype analyzed in AncestryDNA is a proxy for symptomatic influenza that required testing or medical follow-up (ie. with more severe symptoms).

We now explicitly address this important issue throughout the manuscript, including in the new caveats section.

2. It's no wonder there are no GWAS regions reported for flu so far and an analysis of 10 million common variants in 18,334 cases with a positive test for influenza and 276,295 controls is unlikely to help. The authors conclude that "the genetic architectures of COVID-19 and influenza are largely distinct," which is greatly influenced by the selection of controls.

Response: We strongly disagree here. The most likely explanation for the lack of reproducible GWAS discoveries influenza to date is lack of power due to a combination of (i) noisy phenotypes and/or (ii) inadequate sample sizes. Our study addressed the latter, which was sufficient to identify clear and reproducible associations with influenza. With the caveat that interpreting associations with noisy influenza phenotypes is challenging, but we tried to address this in this revision by including results for objective measures of recent influenza infection (please see new Figure 2, which is included below). We also repeated the association analysis in AncestryDNA after restricting the control group to individuals who reported being tested for influenza but had a negative result. Effect sizes for both the ST6GAL1 and B3GALT5 variants remained mostly unchanged (see below).

3. The analyses of rare variants were affected by the same ascertainment issues. This issue is not even brought up in the discussion. The authors picked a very difficult/unreliable phenotype for their analysis and the negative results are what could be expected.

Response: We agree that the same issues discussed above affect associations with rare variants. However, given that we show that a phenotype defined based on a lifetime medical record of influenza is able to identify genome-wide significant associations with common variants (more powered), then we argue that the most likely explanation for the lack of genome-wide significant associations with rare variants (less powered) is lack of power due to sample size.

3. The associations between flu and common variants in ST6GAL1 and B3GALT5 are the only new, very limited and still preliminary data.

Response: We would not describe the associations with ST6GAL1 and B3GALT5 as “preliminary data”, given the very clear replication in independent cohorts.

4. Only transient siRNA-KD for ST6GAL1 resulted in some moderate effect on flu infectivity in vitro. Figure 2 shows a noticeable effect of GAPDH siRNA on ST6GAL1 expression. Similarly, Fig S13 shows an effect of GB3GALT siRNA on GAPDH expression. All significant p-values should be on the plots. Perhaps more optimization of these experiments is required and the effects

of these siRNAs on protein expression of the targets need to be demonstrated by Western blots.

Response: We have added significant P-values to the plots as requested. Yes, the GAPDH control siRNA reduced ST6GAL1 expression in experiment 2 (see Extended Data Fig 7), but this was not statistically significant ($P=0.10$, $N=3$).

As requested, we used Western blots to confirm that ST6GAL1 siRNAs impact protein levels, please see figure below, which is now included in Extended Data Fig 8.

Lastly, we would like to point out that a study published in 2014 also reported that siRNA inhibition of ST6GAL1 reduces influenza infectivity in vitro (<https://pubmed.ncbi.nlm.nih.gov/24670114/>). This study provides independent validation for the functional effect of siRNA inhibition on influenza infectivity observed in our experiments.

5. The effects of putatively associated genetic variants on corresponding proteins and on flu infection should be demonstrated by multiple additional methods. For example, estimating in-vitro infectivity in primary epithelial cells from individuals with different genotypes or in primary cells CRISPR-edited for these genotypes.

Response: Both suggestions are very interesting. However, unfortunately both were either unfeasible or difficult to justify, for the following reasons:

- Comparing infectivity in primary cells from individuals with different genotypes. To perform this experiment, we would need primary cells from a handful of individuals homozygote for the reference and the protective allele. To determine if this was feasible,

we approached a supplier of primary epithelial cells (Lonza) but found that they only have cells available for 8 donors of European ancestry; given that the protective allele has a frequency of 9% in Europeans (even lower in non-Europeans, eg. 3% in African ancestry), we would expect only one individual to be heterozygote and none to be homozygote. So this experiment was not feasible for us to perform.

- Comparing infectivity in primary cells CRISPR-edited for associated genotypes.
 - From our perspective, this experiment is useful to determine which variant is causal at this locus, but it's not necessarily more informative than siRNA inhibition to understand if blocking gene expression influences influenza infectivity.
 - Nonetheless, to perform the experiment suggested by the reviewer, we would need to know which variants are likely causal at this locus, to specifically edit these variants one at a time in primary cells. There are at least 4 candidate causal variants at each locus (variants with $r^2 > 0.8$ with lead variant, see Extended Data Figure 3), and so we would have to perform at least four experiments per locus. Although possible, we felt that this would be a substantial amount of work that was not justifiable given the clear effect of anti-ST6GAL1 siRNAs on influenza infectivity, both in our data and in the published study mentioned above.

6. *Figure 1. rs numbers would be very informative on the plot.*

Response: Updated as requested.

Decision Letter, first revision:

23rd June 2023

Dear Manuel,

Your revised Article "Genetic risk factors for COVID-19 and influenza are largely distinct" has been seen by two of the original referees. You will see from their comments below that, while they find the study improved, they have highlighted a few ongoing concerns. We remain interested in the possibility of publishing your study in Nature Genetics, but we would like to consider your response to these ongoing concerns in the form of a further revision before we make a final decision on publication.

As before, to guide the scope of the revisions, the editors discuss the referee reports in detail within the team, including with the chief editor, with a view to identifying key priorities that should be addressed in revision, and sometimes overruling referee requests that are deemed beyond the scope of the current study. In this case, we ask that you revise the presentation carefully in light of the referees' comments, qualifying the interpretations where needed and addressing the remaining technical points. We again hope that you will find this prioritized set of referee points to be useful

when revising your study. Please do not hesitate to get in touch if you would like to discuss these issues further.

We therefore invite you to revise your manuscript again taking into account all reviewer and editor comments. Please highlight all changes in the manuscript text file. At this stage, we will need you to upload a copy of the manuscript in MS Word .docx or similar editable format.

*2) If you have not done so already, please begin to revise your manuscript so that it conforms to our Article format instructions, available here. Refer also to any guidelines provided in this letter.

Please be aware of our guidelines on digital image standards.

[redacted]

We hope to receive your revised manuscript within 4-8 weeks. If you cannot send it within this time, please let us know.

Nature Genetics is committed to improving transparency in authorship. As part of our efforts in this direction, we are now requesting that all authors identified as 'corresponding author' on published papers create and link their Open Researcher and Contributor Identifier (ORCID) with their account on the Manuscript Tracking System (MTS), prior to acceptance. ORCID helps the scientific community

achieve unambiguous attribution of all scholarly contributions. You can create and link your ORCID from the home page of the MTS by clicking on 'Modify my Springer Nature account'. For more information please visit please visit www.springernature.com/orcid.

Sincerely,
Kyle

Kyle Vogan, PhD
Senior Editor
Nature Genetics
<https://orcid.org/0000-0001-9565-9665>

Referee expertise:

Referee #1: Genome-wide association, infectious diseases, population genetics

Referee #3: Genome-wide association, infectious diseases, virology

Reviewers' Comments:

Reviewer #1:
Remarks to the Author:

I thank the authors for their work to streamline and improve the manuscript. I found it much more impactful and clearer upon revision. However, I have several remaining comments/concerns.

1. The paragraph added at the end of the main text cites the important limitation being that the control group may not have been tested for influenza (page 9-10, lines 184-198). While this is a limitation, another important limitation for the control definition is that it is unlikely that all of the controls were even exposed to influenza in the first place, given the low levels of circulating flu during the years in which this study takes place. The sensitivity analyses cannot get at this limitation, even more of a reason to list it among important limitations in how the interpretation is conducted. This is mentioned in other parts of the text (thank you!), such as page 3-4, lines 52-55. However, that language does not persist throughout the manuscript with many parts still unchanged, including calling the loci "bona fide risk factors for influenza infection" (page 4, line 68) when that may not be appropriate. This is repeated again on page 7, lines 129-130 with "Collectively, these findings establish both loci as bona fide genetic risk factors for influenza infection".

2. The authors should refer to the recent NASEM report on population descriptors when referring to genetic ancestry groups and change their language accordingly when possible. They currently mix ethnic designations (Hispanic) with geographic designations (South Asian). Additionally, it is not appropriate to reassign admixed individuals according to a hierarchy, especially if you are using ethnic

labels which inherently will include admixed individuals. It is especially not appropriate to do so within a hierarchy, which includes picking every non-EUR group over EUR (which reiterates concepts of hypodescent). I realize that this is a little late in the game to change this, but it would be good to have an idea of how many people were actually dropped and reassigned based on this criteria. Additionally, it is not well justified why this stratification was conducted in the first place if all results were pooled later on and there are approaches that would account for substructure without the need for this stratification.

3. Please add p-values or some measure of precision to the genetic correlation in the main text (page 5, line 87), as well as clarify the phenotypes. Right now there is a modest genetic correlation, but if you go to Supp Table 5 the AncestryDNA results have an r_g of 0.211 for infection with a p-value of 0.127. Or if the authors are referring to the EHR influenza meta-analysis infection outcomes, please state that with the actual values ($r_g=0.343$, $p=0.0065$). The genetic correlations in Supp Table 5 all ask different questions and some clarity would be appreciated in the main text. For example, "genetic correlation between these two diseases" on page 5, lines 85-86, is not clear. However, disease and infection are different phenotypes. Someone can be infected and not have the disease, so clarity is needed in language. Supplementary Table 5 has several such comparisons and it is unclear which ones are most relevant or being referred to in this text.

4. I understand that there is limited space, but I think it would benefit the paper to move some of the details in the supplemental information into the main text if possible. Specifically, the ABO association which is discussed in the supplement should be at least outlined in the main text.

5. I still think that it is interesting that the direction of effect for covid versus influenza infection are often opposite. This may be a result of sampling procedures in which those who would be tested for flu were also likely tested for covid and therefore would be opposite samples. (If you had symptoms, you would have either a positive flu test or covid test, but likely not both unless you are exceptionally unlucky.) This is not as striking for severity comparisons (Figure 1), which reinforces that possible bias. This may be less apparent if you stratified by the season (2019-2020 vs 2020-2021), as this bias may not be as strong in 2019-2020 before people were being tested for covid with flu-like symptoms.

Reviewer #3:
Remarks to the Author:

I appreciate the efforts by the authors to address the comments and revise this complicated paper.

The main goal of this effort was to see if the genetic risk factors are the same for COVID-19 and influenza vs. being negative or untested for both conditions during 2019-20 or 2020-21 seasons.

Notably, during this time there was a higher alert for any respiratory infection and increased testing overall (is it COVID or flu or RSV, etc.)? Many flu tests were likely used for differential diagnostics with COVID, not based on disease severity alone.

Anyway, this question has been answered – there are no shared genetic risk factors detectable in this study. Does this mean anything? Unsure.

The approach extremely flexible to the phenotype makes it easy to utilize very large existing datasets but this also sets a very dangerous example of epidemiological analysis of common and recurrent/seasonal infectious diseases where the rules and covariates are optional, and this is not well discussed.

I recommend reading the paper PMID:36649706 and incorporating it into the discussion. It clearly shows the extremely increased chance of spurious associations for infectious diseases, with the bias being proportional to the infection rates.

Four genes were prioritized as effectors: ST6GAL1 and ADIPOQ at the 3q27.3 locus and B3GALT5 and IGSF5 at the 21q22.2. Based on rare missense variants in IGSF5, it was declared the effector gene. Nonetheless, the very limited functional analysis was focused on ST6GAL1 and B3GALT5 without a good explanation why and the other two genes were just mentioned in the supplement.

Extended Data Figures 7 and 9 show plots and p-values combined from 2 individual experiments, which is unusual. Please provide p-values for individual experiments. The fact that target and GAPDH siRNAs mutually affect each other's expression compared to siRNA control is still not good (the p-values here are less informative than the % of decrease).

Ln. 199. In conclusion, we demonstrated the genetic architectures of COVID-19 and influenza are mostly distinct, with few (yet to be identified) shared common genetic risk factors. – What does "with few (yet to be identified)" mean? Looks like an unnecessary statement.

Extended Data Table 1 - can this be provided in a more traditional way, with allele/genotype frequencies in cases and controls and corresponding ORs?

Author Rebuttal, first revision:

Reviewer #1

1. The paragraph added at the end of the main text cites the important limitation being that the control group may not have been tested for influenza (page 9-10, lines 184-198). While this is a limitation, another important limitation for the control definition is that it is unlikely that all of the controls were even exposed to influenza in the first place, given the low levels of circulating flu during the years in which this study takes place. The sensitivity analyses cannot get at this limitation, even more of a reason to list it among important limitations in how the interpretation is conducted. This is mentioned in other parts of the text (thank you!), such as page 3-4, lines 52-55. However, that language does not persist throughout the manuscript with many parts still unchanged, including calling the loci "bona fide risk factors for influenza infection" (page 4, line 68) when that may not be appropriate. This is repeated again on page 7, lines 129-130 with "Collectively, these findings establish both loci as bona fide genetic risk factors for influenza infection".

We appreciate the comments and expanded the discussion to include the confounding aspect of influenza exposure (or lack of it). This is especially relevant for the AncestryDNA cohort that was largely assessed during the pandemic when public health measures limited the spread of influenza. We have also replaced ‘bona fide’ with ‘reproducible’.

2. The authors should refer to the recent NASEM report on population descriptors when referring to genetic ancestry groups and change their language accordingly when possible. They currently mix ethnic designations (Hispanic) with geographic designations (South Asian). Additionally, it is not appropriate to reassign admixed individuals according to a hierarchy, especially if you are using ethnic labels which inherently will include admixed individuals. It is especially not appropriate to do so within a hierarchy, which includes picking every non-EUR group over EUR (which reiterates concepts of hypodescent). I realize that this is a little late in the game to change this, but it would be good to have an idea of how many people were actually dropped and reassigned based on this criteria. Additionally, it is not well justified why this stratification was conducted in the first place if all results were pooled later on and there are approaches that would account for substructure without the need for this stratification.

Thank you for these suggestions. We had originally followed the recommendation of Morales et al., 2018 (PMID: 29448949) to use “Hispanic/Latin American” to describe individuals previously labeled “Admixed American (AMR)” based on earlier nomenclature used by the 1000 Genomes Project. However, as highlighted by the reviewer, the more recent NASEM report recommends against mixing ethnic and geographic designations and so we have now renamed “Hispanic/Latin American (HLA)” as “Admixed American (AMR)”.

We also agree that the criteria used to assign individuals into one ancestry when multiple ancestries had >0.3 probability is arbitrary. Nonetheless, this affected only 0.43% of individuals, as we now detail in the methods section.

Lastly, we do often perform multi-ancestry mega-analyses (ie. a single analysis with all individuals included, irrespective of their ancestry assignments) with our tool Regenie, which adequately accounts for population substructure, as the reviewer alludes to. In our experience, results are almost identical between a multi-ancestry mega-analysis and cross-ancestry meta-analysis. Differences only tend to arise when a substantial fraction of individuals is not assigned to any of the major ancestry groups included in the meta-analysis (these individuals would be included in the mega- but not meta-analysis), which is very rare. For this project, we opted for using ancestry-specific analyses followed by meta-analyses so that we could readily report and compare SNP effect sizes between ancestral groups.

3. Please add p-values or some measure of precision to the genetic correlation in the main text (page 5, line 87), as well as clarify the phenotypes. Right now there is a modest genetic correlation, but if you go to Supp Table 5 the AncestryDNA results have an rg of 0.211 for infection with a p-

value of 0.127. Or if the authors are referring to the EHR influenza meta-analysis infection outcomes, please state that with the actual values ($r_g=0.343$, $p=0.0065$). The genetic correlations in Supp Table 5 all ask different questions and some clarity would be appreciated in the main text. For example, "genetic correlation between these two diseases" on page 5, lines 85-86, is not clear. However, disease and infection are different phenotypes. Someone can be infected and not have the disease, so clarity is needed in language. Supplementary Table 5 has several such comparisons and it is unclear which ones are most relevant or being referred to in this text.

Done as requested (page 6).

4. I understand that there is limited space, but I think it would benefit the paper to move some of the details in the supplemental information into the main text if possible. Specifically, the ABO association which is discussed in the supplement should be at least outlined in the main text.

Done as requested (page 6).

5. I still think that it is interesting that the direction of effect for covid versus influenza infection are often opposite. This may be a result of sampling procedures in which those who would be tested for flu were also likely tested for covid and therefore would be opposite samples. (If you had symptoms, you would have either a positive flu test or covid test, but likely not both unless you are exceptionally unlucky.) This is not as striking for severity comparisons (Figure 1), which reinforces that possible bias. This may be less apparent if you stratified by the season (2019-2020 vs 2020-2021), as this bias may not be as strong in 2019-2020 before people were being tested for covid with flu-like symptoms.

Of the 24 variants reported for COVID-19 hospitalization (16 variants) or SARS-CoV-2 infection (8 variants), for 14 the direction of effect for influenza is opposite that for COVID-19 (see Figure 1). This fraction (14/24 or 58%) does not significantly deviate from 50% (binomial test $P=0.153$), which would be expected under the null hypothesis that all 24 variants are not associated with influenza.

When specifically concentrating on the 8 variants reported for SARS-CoV-2 infection, there is a statistically significant excess of associations that have the opposite direction of effect on reported influenza infection: 7 out of 8, or 88% (binomial test $P=0.004$). To test if this could be partly explained by sampling bias during the COVID-19 pandemic, as suggested by the reviewer, we repeated our reported influenza infection GWAS in the AncestryDNA cohort but this time considering only the response to the question: "The 2019-2020 flu season spans from fall 2019 to late spring 2020. Have you had a flu test in the 2019-2020 flu season?". Individuals who responded "Yes, and I tested positive" were included as cases ($N=3464$). Individuals who responded either "Yes, and I tested negative" or "No" were included as controls ($N=124,914$). In this analysis – denoted "Influenza (2019)" in the plot below – 5 of the 8 (63%) variants had an opposite direction

of effect on reported influenza infection relative to SARS-CoV-2 infection, which was not significantly different from 50% (binomial test $P=0.144$). These results suggest that sampling bias may have partly contributed to the observation that variants associated with SARS-CoV-2 infection have an opposing effect on reported influenza infection. However, we stress that aside from the *ABO* locus, the association between these 8 variants and reported influenza infection is very weak, and therefore may simply represent false-positive associations with effect sizes that randomly fluctuate around $OR=1$.

Reviewer #3

1. I recommend reading the paper PMID:36649706 and incorporating it into the discussion. It clearly shows the extremely increased chance of spurious associations for infectious diseases, with the bias being proportional to the infection rates.

We read with interest the paper the reviewer mentions above, and incorporated its findings into the discussion. This paper makes the following major observations:

- Using simulated data, the authors found that when a variant X affects risk of exposure to a pathogen but not risk of infection once exposed, then:
 - Using population-level controls (as opposed to exposed but without symptomatic infection) can sometimes yield a spurious association between variant X and risk of infection (see their Figure 2)
 - Intuitively, this makes sense: 100% of individuals with symptomatic infection had pathogen exposure, whereas that is unlikely to be the case for 100% of population-level controls. In other words, the outcome that is determined by variant X (risk of exposure) will often be more common in individuals with symptomatic infection than in population-level controls.
 - The probability that variant X will show a spurious association with risk of infection when using population-level controls decreases with increasing prevalence of pathogen exposure (see their Table 2).
 - This also makes sense intuitively: at the extreme, when the prevalence of pathogen exposure is 100%, the outcome that is determined by variant X (risk of exposure) has the same frequency in both the case and control groups, and so no association is observed with variant X.
- Using real world data, the authors compared results between two GWAS that used the same set of cases but different control groups:
 - GWAS #1: 702 individuals infected with hepatitis C virus (HCV) but who cleared the virus (source: HCV consortium) vs. 1,037 individuals with persistent HCV infection (source: HCV consortium). Two loci at $P < 5 \times 10^{-8}$: *HLA* and *IFNL3*.
 - GWAS #2: 702 individuals infected with hepatitis C virus (HCV) but who cleared the virus (source: HCV consortium) vs. 370K ancestry-matched population-level controls with unknown HCV status (source: UKB). This GWAS identifies the same two loci discovered in GWAS #1 (*HLA*, *IFNL3*) and two additional loci at $P < 5 \times 10^{-8}$: *STX18* and a locus on chr 2 near *MIR3681HG*.
- Additional analyses suggest that one of the novel associations discovered in GWAS #2 (*STX18*) is an example of a locus that affects risk of exposure to HCV and not risk of HCV clearance. So the authors suggest that the *STX18* variant increases risk of hemophilia, which in turn increases risk of exposure to HCV. This conclusion is based on the following observations:

- The signal was attenuated (but not fully eliminated) after excluding individuals with hemophilia.
- The signal was considerably attenuated and not significant in GWAS #1;
- The signal was stronger when comparing all HCV cases (with cleared or persistent infection) against UKB population-level controls.

If the type of bias described in this paper explained the two associations we discovered for influenza (*ST6GAL1* and *B3GALT5*), then these loci would have to be strongly associated with an outcome that increased the risk of exposure to the influenza virus. Unlike HCV, it is not obvious what heritable outcome might strongly increase risk of exposure to the influenza virus. Nonetheless, to address this possibility, we performed a phenome-wide association study (PheWAS) for the *ST6GAL1* and *B3GALT5* variants in the UKB (N=450K) and GHS (N=175K) studies, testing each variant for association with 6221 binary outcomes and 2916 quantitative outcomes. We found no outcomes associated with either variant at a $P < 0.05 / (2 \text{ loci} * 9137 \text{ traits tested}) = 2.68 \times 10^{-6}$, indicating that these variants do not have a strong effect on any outcome measured in these studies. As such, we think it is highly unlikely that the associations we describe for influenza arise from the confounding effect of a correlated outcome.

We have added a sentence to the discussion to highlight this potential confounder effect and a summary of the PheWAS results described above, which are described in the Supplementary Information.

2. Four genes were prioritized as effectors: ST6GAL1 and ADIPOQ at the 3q27.3 locus and B3GALT5 and IGSF5 at the 21q22.2. Based on rare missense variants in IGSF5, it was declared the effector gene. Nonetheless, the very limited functional analysis was focused on ST6GAL1 and B3GALT5 without a good explanation why and the other two genes were just mentioned in the supplement.

Apologies if it wasn't clear why we selected only *ST6GAL1* and *B3GALT5* (out of the 4 likely effector genes) for functional experiments. We have now edited the beginning of this section to explain our rationale more clearly:

“Lastly, we performed in vitro experiments to study the impact of gene expression knockdown on influenza virus H1N1 (Puerto Rico 8 strain) infectivity. For these experiments, we focused on two – *ST6GAL1* and *B3GALT5* – of the four likely effector genes of influenza-associated variants because of their potential role in a critical step of influenza virus infectivity, modulation of α -2,6-linked sialic acid abundance at the cell surface.”

And a minor correction, we did not declare *IGSF5* as “the effector gene” at the 21q22.2 locus. We stated that results from exome sequencing provided additional support for this gene being an effector gene at this locus. But perhaps this was not clear, so we edited that section to read:

“This observation provides additional support for IGSF5 being one of the likely effector genes underlying the common variant association with flu at the 21q22.2 locus.”

3. *Extended Data Figures 7 and 9 show plots and p-values combined from 2 individual experiments, which is unusual. Please provide p-values for individual experiments. The fact that target and GAPDH siRNAs mutually affect each other's expression compared to siRNA control is still not good (the p-values here are less informative than the % of decrease).*

Done as requested.

4. *Ln. 199. In conclusion, we demonstrated the genetic architectures of COVID-19 and influenza are mostly distinct, with few (yet to be identified) shared common genetic risk factors. – What does “with few (yet to be identified)” mean? Looks like an unnecessary statement.*

We have removed the “(yet to be identified)” statement.

5. *Extended Data Table 1 - can this be provided in a more traditional way, with allele/genotype frequencies in cases and controls and corresponding ORs?*

Done as requested.

Decision Letter, second revision:

7th December 2023

Dear Manuel,

Your revised manuscript "Genetic risk factors for COVID-19 and influenza are largely distinct" (NG-LE60433R2) has been seen by two of the original referees. As you will see from their comments below, they find that the paper has improved in revision, and therefore we will be happy in principle to publish it in Nature Genetics as a Letter pending final revisions to address the referees' remaining requests and to comply with our editorial and formatting guidelines.

We are now performing detailed checks on your paper, and we will send you a checklist detailing our editorial and formatting requirements soon. Please do not upload the final materials or make any revisions until you receive this additional information from us.

Thank you again for your interest in Nature Genetics. Please do not hesitate to contact me if you have any questions.

Sincerely,
Kyle

Kyle Vogan, PhD
Senior Editor
Nature Genetics
<https://orcid.org/0000-0001-9565-9665>

Reviewer #1 (Remarks to the Author):

Thank you to the authors for their revisions. The majority of my comments have been addressed.

I would like to point out that while the NASEM report discourages the mixing of racial and ethnic terms with genetic terms, it does not advocate for the use of "Admixed American", and in fact calls it out as an inappropriate term as it is not specific. In addition, I would suggest the authors change their terminology as indicated in the report to be more specific to their reference data, such as "1kG-like AFR", etc.

Besides this, which could be solved with some additional terminology/acronyms, I have no remaining comments to be addressed.

Reviewer #3 (Remarks to the Author):

I appreciate the efforts of the authors to push the envelope of statistical analysis of infectious diseases.

This is not an easy task even though the authors engaged all the resources in the world.

There are some additional points to consider:

1. P4. Ln 41. "This study identified 24 independent genetic risk factors for COVID-19, suggesting a role for 33 genes in disease etiology, including MUC5B, SFTPD and SLC22A31, among others". Why are these specific genes mentioned? There is no obvious connection to the content of the paper.

2. Having a negative flu test during 2019-20 or 2020-21 seasons is presented as a better control compared to not having a test at all. Having a one-time negative test for flu during several highly masked years and isolation is not much more informative than not having a test. This should be pointed out, including in study limitations such as:

An important limitation of our study is that the control group in the discovery and replication influenza GWAS may have included individuals (i) infected with influenza but that were asymptomatic or never tested; and (ii) who tested negative because they were not exposed to the influenza virus.

3. Rs16861415-A is listed as an effect allele in Extended Data Table 1 with an effect allele frequency of 0.064 vs 0.074 and 0.070 vs 0.079 in Discovery and Replication studies.

This must be a mistake as Rs16861415-A is the major allele in all populations.

Extended Data Figure 2: Rs16861415-C is now listed correctly but with a meta-analysis allele frequency of 0.097 (including FinnGen, which is mentioned as not available in Extended Data Table 1). The 0.097 is way outside of the frequency ranges reported in both Discovery and Replication cohorts – max 0.074 and 0.079 in controls. AAF – used in Figures should be spelled out.

Extended Data Table 1 lists rs2837112 – C as an effect allele, while figures show rs2837112-A allele.

4. Please consistently include rs numbers in all Supplementary tables (Table 3, 7, 8, 9). Providing effect allele frequencies in comparison groups would also be more informative than just the total allele frequency.

Author Rebuttal, second revision:

Reviewer #1

Thank you to the authors for their revisions. The majority of my comments have been addressed.

1. I would like to point out that while the NASEM report discourages the mixing of racial and ethnic terms with genetic terms, it does not advocate for the use of "Admixed American", and in fact calls it out as an inappropriate term as it is not specific. In addition, I would suggest the authors change their terminology as indicated in the report to be more specific to their reference data, such as "1kG-like AFR", etc.

Besides this, which could be solved with some additional terminology/acronyms, I have no remaining comments to be addressed.

We have aligned the genetic ancestry groups terminology with the NASEM report using as the report and you suggest.

Reviewer #3

1. P4. Ln 41. "This study identified 24 independent genetic risk factors for COVID-19, suggesting a role for 33 genes in disease etiology, including MUC5B, SFTPD and SLC22A31, among others". Why are these specific genes mentioned? There is no obvious connection to the content of the paper.

Thank you for pointing this out. We removed those specific genes such that the sentence now reads: "This study identified 24 independent genetic risk factors for COVID-19."

2. *Having a negative flu test during 2019-20 or 2020-21 seasons is presented as a better control compared to not having a test at all. Having a one-time negative test for flu during several highly masked years and isolation is not much more informative than not having a test. This should be pointed out, including in study limitations such as:*

An important limitation of our study is that the control group in the discovery and replication influenza GWAS may have included individuals (i) infected with influenza but that were asymptomatic or never tested; and (ii) who tested negative because they were not exposed to the influenza virus.

We edited the sentence in question in accordance with the suggestions above.

3a. *Rs16861415-A is listed as an effect allele in Extended Data Table 1 with an effect allele frequency of 0.064 vs 0.074 and 0.070 vs 0.079 in Discovery and Replication studies.*

This must be a mistake as Rs16861415-A is the major allele in all populations.

Thank you for pointing this mistake out. You are correct that A is the major allele and we have fixed this error by reporting the correct effect allele, C, which is the minor allele.

3b. *Extended Data Figure 2: Rs16861415-C is now listed correctly but with a meta-analysis allele frequency of 0.097 (including FinnGen, which is mentioned as not available in Extended Data Table 1). The 0.097 is way outside of the frequency ranges reported in both Discovery and Replication cohorts – max 0.074 and 0.079 in controls. AAF – used in Figures should be spelled out.*

9.7% is actually not outside the allele frequency ranges as it reflects the influence of the elevated Finnish allele frequency of 15.6% in FinnGen. In fact, FinnGen's allele frequency aligns quite well with the Finnish allele frequency for rs16861415:C reported in gnomADv4 (16.4%; https://gnomad.broadinstitute.org/variant/3-186978576-T-C?dataset=gnomad_r4). The issue pointed out by the reviewer arises because we accidentally left out the FinnGen results from Table S6, which has since been corrected. We also updated the figure legends to properly spell out the AAF abbreviation.

3c. *Extended Data Table 1 lists rs2837112 – C as an effect allele, while figures show rs2837112-A allele.*

We corrected this error and now report the correct effect allele (A) in Extended Data Table 1.

4. Please consistently include rs numbers in all Supplementary tables (Table 3, 7, 8, 9). Providing effect

allele frequencies in comparison groups would also be more informative than just the total allele frequency.

Thank you for your suggestion, we have included rsIDs and case/control effect allele frequencies in the supplementary tables.

Final Decision Letter:

24th June 2024

Dear Manuel,

I am delighted to say that your manuscript "Genetic risk factors for COVID-19 and influenza are largely distinct" has been accepted for publication in an upcoming issue of Nature Genetics.

Your paper will be published online after we receive your corrections and will appear in print in the next available issue. You can find out your date of online publication by contacting the Nature Press Office (press@nature.com) after sending your e-proof corrections.

Before your paper is published online, we will be distributing a press release to news organizations worldwide, which may very well include details of your work. We are happy for your institution or funding agency to prepare its own press release, but it must mention the embargo date and Nature Genetics. Our Press Office may contact you closer to the time of publication, but if you or your Press Office have any enquiries in the meantime, please contact press@nature.com.

Please note that Nature Genetics is a Transformative Journal (TJ). Authors may publish their research with us through the traditional subscription access route or make their paper immediately open access through payment of an article-processing charge (APC). Authors will not be required to make a final decision about access to their article until it has been accepted. Find out more about Transformative Journals

Authors may need to take specific actions to achieve compliance with funder and institutional open access mandates. If your research is supported by a funder that requires immediate open access (e.g. according to Plan S principles), then you should select the gold OA route, and we will direct you to the compliant route where possible. For authors selecting the subscription publication route, the journal's standard licensing terms will need to be accepted, including [a href="https://www.nature.com/nature-portfolio/editorial-policies/self-archiving-and-license-to-publish"](https://www.nature.com/nature-portfolio/editorial-policies/self-archiving-and-license-to-publish). Those licensing terms will supersede any other terms that the author or any third party may assert apply to any version of the manuscript.

If you have not already done so, we strongly recommend that you upload the step-by-step protocols used in this manuscript to protocols.io. protocols.io is an open online resource that allows researchers to share their detailed experimental know-how. All uploaded protocols are made freely available and are assigned DOIs for ease of citation. Protocols can be linked to any publications in which they are used and will be linked to from your article. You can also establish a dedicated workspace to collect all your lab Protocols. By uploading your Protocols to protocols.io, you are enabling researchers to more readily reproduce or adapt the methodology you use, as well as increasing the visibility of your protocols and papers. Upload your Protocols at <https://protocols.io>. Further information can be found at <https://www.protocols.io/help/publish-articles>.

Sincerely,
Kyle

Kyle Vogan, PhD
Senior Editor
Nature Genetics
<https://orcid.org/0000-0001-9565-9665>